# On Large Language Model Continual Unlearning

**Chongyang Gao**[1][*][†]**, Lixu Wang**[1][*][†]**, Kaize Ding**[1]**, Chenkai Weng**[2]**, Xiao Wang**[1]**, Qi Zhu**[1]
[1]Northwestern University, [2]Arizona State University
{cygao, lixuwang2025}@u.northwestern.edu, chenkai.weng@asu.edu
{kaize.ding,wangxiao,qzhu}@northwestern.edu

## Abstract

While large language models have demonstrated impressive performance across various domains and tasks, their security issues have become increasingly severe. Machine unlearning has emerged as a representative approach for model safety and security by removing the influence of undesired data on the target model. However, these methods do not sufficiently consider that unlearning requests in real-world scenarios are continuously emerging, especially in the context of LLMs, which may lead to accumulated model utility loss that eventually becomes unacceptable. Moreover, existing LLM unlearning methods often ignore previous data access limitations due to privacy concerns and copyright protection. Without previous data, the utility preservation during unlearning is much harder. To overcome these challenges, we propose the $O^3$ framework that includes an $\underline{O}$rthogonal low-rank adapter (LoRA) for continually unlearning requested data and an $\underline{O}$ut-$\underline{O}$f-Distribution (OOD) detector to measure the similarity between input and unlearning data. The orthogonal LoRA achieves parameter disentanglement among continual unlearning requests. The OOD detector is trained with a novel contrastive entropy loss and utilizes a glocal-aware scoring mechanism. During inference, our $O^3$ framework can decide whether and to what extent to load the unlearning LoRA based on the OOD detector's predicted similarity between the input and the unlearned knowledge. Notably, $O^3$'s effectiveness does not rely on any retained data. We conducted extensive experiments on $O^3$ and state-of-the-art LLM unlearning methods across three tasks and seven datasets. The results indicate that $O^3$ consistently achieves the best unlearning effectiveness and utility preservation, especially when facing continuous unlearning requests. The source codes can be found at https://github.com/GCYZSL/O3-LLM-UNLEARNING.

## 1 Introduction

Recently, bolstered by scaling laws (Kaplan et al., 2020), the size of language models has grown tremendously, demonstrating excellent performance across various tasks (Wang et al., 2024). However, concerns about large language models (LLMs) have also increased, particularly regarding how to eliminate undesirable data influence (*e.g.*, privacy information (Pan et al., 2020)). To address this issue, *machine unlearning* (Bourtoule et al., 2021) is applied in LLMs to remove private, toxic, or illegal data. Current methods for LLM unlearning can be primarily categorized into parameter optimization (Chen & Yang, 2023; Eldan & Russinovich, 2023; Jia et al., 2024; Zhang et al., 2024; Meng et al., 2022; Li et al., 2024), and in-context unlearning (Thaker et al., 2024; Pawelczyk et al., 2024). The parameter optimization methods involve directly fine-tuning the LLM, with the objective typically being to maximize the task loss on the unlearning data or to minimize the random label loss. Some methods identify the related parameters and then make appropriate modifications. In-context learning-based methods modify the LLM input prompts to make the LLM refuse to output content related to the unlearning data. Regarding unlearning effectiveness, parameter optimization is typically much more effective than in-context learning.

However, these methods still often poorly maintain the model utility outside the unlearned knowledge, especially in real-world continual settings. The challenges are two-fold: (**i**): First, in addition

---

[*]Equal contributions (ordered alphabetically); [†]Corresponding author.

to the data that needs to be unlearned, existing unlearning methods also require a large dataset called the *retained dataset* to maintain the model utility. This retained dataset often consists of the original training dataset (Bourtoule et al., 2021) or a portion of it, but as LLMs are trained on massive datasets (Wang et al., 2024), assuming access to the complete training data is typically unrealistic (Liu et al., 2024). Moreover, as time goes on, the original training data of LLMs may become inaccessible due to expired access authorization, data privacy, and intellectual property protection (Sun et al., 2024). If the retained dataset only contains incomplete training data distribution, the model utility of the missing parts significantly declines after unlearning. Although some studies shrink the range of the retained data to the distribution most susceptible to unlearning, this distribution itself is hard to characterize and its data may be limited due to intrinsic rarity and privacy protection (Chang et al., 2024; Huang et al., 2024). (**ii**): The second challenge is that existing LLM unlearning methods only consider single operations and cannot perform effective continual unlearning. LLM unlearning is often not a one-off operation but a continual process, as unlearning requests continuously emerge in the real world (Liu et al., 2024). As the number of unlearning operations increases, the aforementioned decline in model utility will also have a cumulative effect, even with the retained dataset, meaning that the model's general capabilities will significantly decrease Gu et al. (2024); Gupta et al. (2024) over time.

In this work, to achieve more effective continual unlearning for LLMs, we propose the $\bigcirc^3$ framework, which ***can balance unlearning effectiveness and model utility preservation in continuous scenarios without using any retained data.*** At a high level, the $\bigcirc^3$ framework mainly includes an **O**rthogonal Low-rank adapter (LoRA) (Hu et al., 2021) for continuously unlearning requested data and an **O**ut-**O**f-Distribution (OOD) detection module to assess the similarity between input data and unlearning data. Specifically, the orthogonal LoRA in $\bigcirc^3$ enables the disentanglement of parameter space across different unlearning requests, ensuring that the unlearning effectiveness of different requests does not interfere with each other. Then the OOD detector in $\bigcirc^3$ is trained with a novel contrastive entropy loss as its backbone and supplemented with a glocal-aware scoring mechanism. The $\bigcirc^3$ framework can balance unlearning and utility because it smartly leverages the data similarity determined by the OOD detector to decide whether and to what extent to load the unlearning LoRA during inference. In summary, the main contributions of this work include:

- We study the underexplored problem of LLM continual unlearning and tackle the challenge of balancing unlearning effectiveness and model utility preservation when LLM faces the continuous arrival of unlearning requests, without using any retained data.
- We propose a novel $\bigcirc^3$ framework that includes an orthogonal unlearning LoRA and an OOD detector. The orthogonal design of LoRA prevents interference among different unlearning requests, achieving better unlearning effectiveness in continuous scenarios. The OOD detector measures the similarity between input and unlearning data, allowing $\bigcirc^3$ to decide whether and to what extent to load the unlearning LoRA during inference.
- We conduct extensive experiments on multiple benchmark tasks that comprehensively test the LLM continual unlearning on discriminative, generative, and reasoning tasks. The experiment results demonstrate that $\bigcirc^3$ consistently achieves the best balance between unlearning effectiveness and utility preservation without using any retained data, compared with many state-of-the-art baseline methods when facing continuous unlearning requests.

## 2 PRELIMINARY

**Language Model Unlearning.** Machine unlearning (Bourtoule et al., 2021) is proposed to protect data privacy and ensure authorized usage (Liu et al., 2024; Zhang et al., 2023). There have been approaches to achieving unlearning through parameter optimization (Jang et al., 2023; Golatkar et al., 2020; Yao et al., 2023; Eldan & Russinovich, 2023; Zhang et al., 2024; Jia et al., 2024; Meng et al., 2022; Yu et al., 2023; Wu et al., 2023; Li et al., 2024), and in-context learning (Thaker et al., 2024; Pawelczyk et al., 2024). The optimization-based unlearning is to employ GradAsc (Golatkar et al., 2020; Yao et al., 2023) on the unlearned data. The following approaches, like PO (Eldan & Russinovich, 2023; Zhang et al., 2024; Jia et al., 2024), notice that unconstrained GradAsc hurts the model's utility, thus crafting task labels through shuffling or rejection. Yu et al. (2023) localizes the model parameters related to unlearning data and updates them through merging or subtracting (Ding et al., 2023). The in-context learning-based methods adjust input prompts to reject unwanted content generation. Although these approaches can achieve unlearning in certain cases, they neglect that

unlearning requests always appear continuously (Liu et al., 2024). The challenge is that the accumulative impact of conducting unlearning sequentially induces catastrophic model utility loss (Gu et al., 2024; Gupta et al., 2024). Besides, their success in a single unlearning operation relies on sufficient retained data with accurate labels, which are scarce in real-world LLM applications. Our $\bigcirc^3$ addresses the above challenges via orthogonal unlearning LoRA and a designed OOD detector.

**Out-of-distribution Detection.** Unlike traditional OOD detection based on one-class SVM (Erfani et al., 2016), random forest (Primartha & Tama, 2017), and Gaussian mixture modeling (Laxhammar et al., 2009), deep learning-based OOD detection (Yang et al., 2021) has become mainstream, focusing on the classification task. These approaches assume the availability of category-labeled ID data. However, only a few studies explore unsupervised OOD detection in language models with only unlabeled ID data. DAGMM (Zong et al., 2018) uses a deep auto-encoder to generate low-dimensional representations to estimate OOD scores, only working on multivariate time-series data. Xu et al. (2021) leverage the feature extracted from a pre-trained language model and then fit an OC-SVM for detection. More recent related works search ways like ensemble learning (Zhou et al., 2023), pseudo labeling (Lang et al., 2022), outlier exposure (Cao et al., 2024), and prefix tuning (Ouyang et al., 2023) to achieve OOD text intent detection. Although these works can achieve OOD detection over non-classification tasks, their effectiveness still relies on labels of specific tasks. Our proposed $\bigcirc^3$ does not require any types of labels.

**Problem Definition.** In the context of LLM, we consider the prevalent causal language models, which take textual token sequences with variable lengths as the input. As an autoregressive model, the target LLM $M_\Theta$ parameterized on $\Theta$ calculates the probability of each token in a text with the previous tokens. For the problem of unlearning, we suppose a sequence of continually arriving unlearning requests with $N^{\mathrm{U},t}$ data samples, i.e., $\{\mathcal{D}^{\mathrm{U},t}\}_{t=1}^T$ where, for $t$-th request, $\mathcal{D}^{\mathrm{U},t} = \{\boldsymbol{x}_i \| \boldsymbol{x}_i \sim \mathcal{P}_{\mathcal{X}}^{\mathrm{U},t}\}_{i=1}^{N^{\mathrm{U},t}}$ and $\mathcal{P}_{\mathcal{X}}^{\mathrm{U},t}$ is the input marginal distribution. $T$ is an index of the latest request. Each unlearning request corresponds to a task with the training data characterized by the input $\mathcal{P}_{\mathcal{X}}^t$ and label distributions $\mathcal{P}_{\mathcal{Y}}^t$. Traditional unlearning approaches always assume a retained dataset drawn from a distribution $\mathcal{P}_{\mathcal{X}}^{\mathrm{R},t}$ disjoint from the unlearning one $\mathcal{P}_{\mathcal{X}}^{\mathrm{U},t}$, to retain the model performance on the original training distribution. Then, on one hand, the straightforward objective of continual unlearning is:

$$\sum_{t=1}^T \min_{\Theta^t} \mathbf{I}(M_{\boldsymbol{x} \sim \mathcal{P}_{\mathcal{X}}^{\mathrm{U},t}}(\boldsymbol{x}, \Theta); M_{\boldsymbol{x} \sim \mathcal{P}_{\mathcal{X}}^{\mathrm{U},t}}^t(\boldsymbol{x}, \Theta^t)), \sum_{t=1}^T \max_{\Theta^t} \mathbf{I}(M_{\boldsymbol{x} \sim \mathcal{P}_{\mathcal{X}}^{\mathrm{R},t}}(\boldsymbol{x}, \Theta); M_{\boldsymbol{x} \sim \mathcal{P}_{\mathcal{X}}^{\mathrm{R},t}}^t(\boldsymbol{x}, \Theta^t)),$$
(1)

where $M^t$ parameterized on $\Theta^t$ represents the target model during and after unlearning on the $t$-th unlearning set $\mathcal{D}^{\mathrm{U},t}$, and $\mathbf{I}(\cdot; \cdot)$ calculates the mutual information between two random variables. On the other hand, the model utility preservation on other distributions $\mathcal{P}_{\mathcal{X}}^{\mathrm{O}}$ distinct from the unlearning one, is another important objective for unlearning,

$$\sum_{t=1}^T \max_{\Theta^t} \mathbf{I}(M_{\boldsymbol{x} \sim \mathcal{P}_{\mathcal{X}}^{\mathrm{O}}}(\boldsymbol{x}, \Theta); M_{\boldsymbol{x} \sim \mathcal{P}_{\mathcal{X}}^{\mathrm{O}}}^t(\boldsymbol{x}, \Theta^t)).$$
(2)

## 3 METHODOLOGY

**Proposed Approach Overview.** To handle continual unlearning requests for LLM, especially considering the constraints in acquiring the retained dataset, we propose a framework called $\bigcirc^3$ that contains two major modules: an **O**rthogonal LoRA module to unlearn requested knowledge continually and an **OO**D detector-like module to detect unlearning knowledge, detailed in Sections 3.1 and 3.2, respectively. $\bigcirc^3$ continuously unlearns the requested data using LoRA with an orthogonal regularization loss that can maintain continuous unlearning performance. $\bigcirc^3$ obtains the unlearning knowledge detection module by the contrastive entropy minimization and local-global layer-aggregated scoring techniques, which can predict the probability that the input data sample belongs to the unlearning distribution $\mathcal{P}_{\mathcal{X}}^{\mathrm{U},t}$. These two major modules only use the unlearning dataset of each unlearning request and do not require the access of retained data. After unlearning, $\bigcirc^3$ works with an effective inference mechanism (Section 3.3), in which the unlearning LoRA is loaded with soft weights originating from the probability predicted by the OOD module to produce distinct outputs for different data.

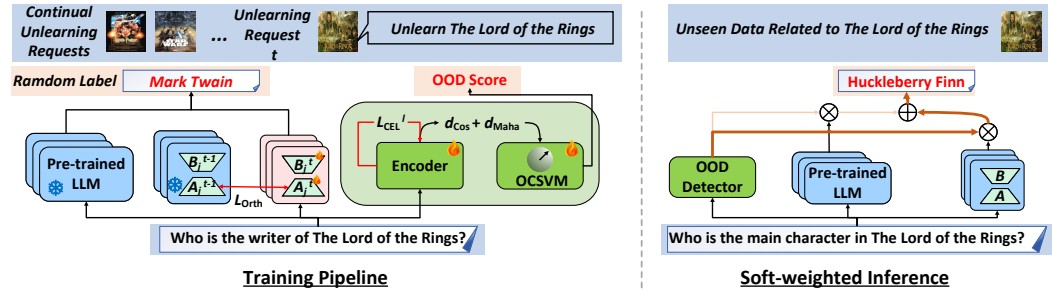

Figure 1: The overview of $\circ^3$ framework to handle continual unlearning requests for LLM without using any retained data. $\circ^3$ includes two major components: an Orthogonal optimization process for unlearning requested knowledge, and an OOD detector is used to detect whether the input contains the unlearning knowledge. The unlearning knowledge optimization uses the orthogonal loss ($\mathcal{L}_{\mathrm{Orth}}$) to prevent interference among different unlearning requests. The OOD detector is trained by a novel contrastive entropy loss ($\mathcal{L}_{\mathrm{CEL}}$) and works with a layer-aggregated scoring mechanism that leverages cosine similarity ($\mathrm{d}_{\mathrm{Cos}}$) and Mahalanobis distance ($\mathrm{d}_{\mathrm{Maha}}$). In the inference phase, the OOD detector decides whether and to what extent to load the unlearning LoRA.

### 3.1 LLM Unlearning with Orthogonal-Regularized LoRA

As the model size of LLM is quite large, the associated costs of fine-tuning-based unlearning are extremely massive. Moreover, continual fine-tuning of the entire LLM also poses challenges of preserving the general capabilities and sustaining the efficacy of previous unlearning efforts (Gu et al., 2024). To address these issues, we use LoRA and incorporate an orthogonal regularization by keeping the original LLM frozen and focusing on updating previously neglected LoRA parameters.

LoRA (Hu et al., 2021) employs low-rank matrix decomposition to reduce the trainable parameters. A transformer's attention or linear layers are parameterized on weight matrices, $W \in \mathbb{R}^{U \times V}$. LoRA introduces two low-rank trainable matrices $r := \{A, B\}$ where $A \in \mathbb{R}^{U \times K}$ and $B \in \mathbb{R}^{K \times V}$. This ensures that the dimension of $AB\boldsymbol{h}$ matches that of $W\boldsymbol{h}$ where $\boldsymbol{h}$ is the input of the weight matrix,

$$\boldsymbol{h}' = W\boldsymbol{h} + \triangle W\boldsymbol{h} = W\boldsymbol{h} + AB\boldsymbol{h}, \tag{3}$$

where $\boldsymbol{h}'$ is the output of the weight matrix. During LoRA fine-tuning, $W$ is frozen, and only $A$ and $B$ are updated. To achieve effective unlearning, we leverage preference optimization to update the model for random task labels or reject-based answers such as 'I don't know', termed as $\hat{y}$. We adopt CrossEntropy Loss of the unlearning set $\mathcal{D}^{\mathrm{U},t}$ for training,

$$\mathcal{L}_{\mathrm{CE}}^{t} = -\frac{1}{N^{\mathrm{U},t}} \sum_{i=1}^{N^{\mathrm{U},t}} \hat{y}_{i}^{\mathrm{U},t} \log M_{\Theta}(\boldsymbol{x}_{i}^{\mathrm{U},t}). \tag{4}$$

To further reduce the cost during unlearning, we leverage a single LoRA for all unlearning tasks rather than employing multiple LoRAs. However, this adoption compromises the effectiveness of previously unlearned requests due to task entanglement. To address such issues, inspired by Wang et al. (2023), we propose to adopt an orthogonal regularization loss to disentangle unlearning requests in the LoRA parameter space. Specifically, at the $t$-th unlearning request, we initialize the LoRA with the latest $r^{t-1} := \{A^{t-1}, B^{t-1}\}$. The matrix $A_{t-1}$ can be further denoted as $[\boldsymbol{a}^{t-1,1}, \cdots, \boldsymbol{a}^{t-1,K}]$ where each $\boldsymbol{a}^{t-1,k}$ is a column vector with $U$ dimensions. Note that the matrix $B$ can be regarded as the linear weights of matrix $A$. Thus, we don't consider its interference among different requests. We view the spanned space that consists of column vectors in $A^t$ as the $t$-th task parameter space $\mathcal{U}^t$. In this case, if the LoRA parameter space of different requests achieves orthogonal, their spanned space will be:

$$\forall \boldsymbol{u} \in \mathcal{U}^t, \forall \boldsymbol{v} \in \mathcal{U}^{t-1}, \boldsymbol{u} \cdot \boldsymbol{v} = 0. \tag{5}$$

This design guides the LoRA tuning to focus more on previously overlooked parameters, striking a balance between different unlearning tasks. Therefore, we can approximate this correlation with an orthogonal loss $\mathcal{L}_{\mathrm{Orth}}$ and combine it with the preference optimization to form the total loss of the LoRA tuning $\mathcal{L}_{\mathrm{LoRA}}$,

$$\mathcal{L}_{\mathrm{Orth}}^{t} = \left\| (A^{t-1})^{\top} A^t \right\|^2, \text{and } \mathcal{L}_{\mathrm{LoRA}}^{t} = \mathcal{L}_{\mathrm{CE}}^{t} + \lambda \mathcal{L}_{\mathrm{Orth}}^{t}, \tag{6}$$

where $\lambda = 0.1$ is a factor used to balance the optimization guidance, and please refer to its sensitivity analysis in Section 4.3. Initially, when $t = 1$, there is no need to apply the orthogonal regularization. As the unlearning continues, the orthogonal loss is incorporated.

## 3.2 UNLEARNED KNOWLEDGE DETECTION

By viewing the unlearning dataset as in-distribution (ID) data, our task of unlearning knowledge detection becomes an OOD detection problem (Yang et al., 2021). However, nearly all existing OOD detection works are built upon the classification problem, which is not the mainstream task of language models. Besides, their representation learning and scoring mechanisms rely on semantic category labels that are inaccessible in our problem. To tackle these shortcomings, we propose new designs for the representation learning and scoring mechanism for LLM continual unlearning.

**OOD Representation Learning with Contrastive Entropy Minimization.** In unsupervised scenarios, self-supervised learning (SSL), such as contrastive learning, has been demonstrated effectively for OOD detection on visual data (Tack et al., 2020). However, such contrastive learning is unsuitable for language OOD detection, as it is hard to achieve semantically equivalent data augmentation on texts. Moreover, token-level SSL tasks, like Masked Language Modeling (MLM) and SimCSE (Gao et al., 2021; Jian et al., 2022a;b), are far less effective. To address these challenges, we propose a novel contrastive entropy loss to learn text representations for OOD detection.

Our contrastive entropy also starts with the augmentation view generation. Inspired by MLM, we leverage random masking to generate the first view type. Specifically, for a particular text instance $\boldsymbol{x}$ with the token length $n$, we randomly select $p\%$ ($p = 15$ in our implementation) tokens and replace them with the token of $[\mathrm{MASK}]$. The instance with the random masking is denoted as $\boldsymbol{x}^*$. For the second contrastive view, we follow MoCo (He et al., 2020) to maintain a key encoder $F_{\Omega^{\mathrm{key}}}$ initialized from the original OOD module backbone $F_{\Omega}$ that is a transformer consisting of $L$ attention layers, i.e., $F := f_{\omega_1} \circ \cdots f_{\omega_l} \circ \cdots f_{\omega_L}$. Then the second view is generated from $F_{\Omega^{\mathrm{key}}}$ by forwarding the original text instance $\boldsymbol{x}$.

Our contrastive entropy loss shares similar intuition with the standard contrastive learning, i.e., aligning positive pairs as close as possible while pushing negative pairs far away, but converges much faster owing to weighting by cosine similarity-based Softmax probability and being conducted at every model layer. In the mini-batch training, our contrastive entropy loss shapes like

$$\mathcal{L}_{\mathrm{CEL}} = -\sum_{i=1}^{N^B} \sum_{l=1}^{L} \sum_{j=1}^{N^B} \Delta(i,l,j) \log(\Delta(i,l,j)), \text{where } \Delta(i,l,j) = \frac{\exp(f_{\omega_{[1:l]}}(\boldsymbol{x}_i^*) \cdot f_{\omega_{[1:l]}^{\mathrm{key}}}(\boldsymbol{x}_j))}{\sum_{k=1}^{N^B} \exp(f_{\omega_{[1:l]}}(\boldsymbol{x}_i^*) \cdot f_{\omega_{[1:l]}^{\mathrm{key}}}(\boldsymbol{x}_k))},$$

(7)

where $f_{\omega_{[1:l]}}(\boldsymbol{x}_i^*)$ means the token averaging representation of the $l$-th layer. $N^B$ is the sample quantity of a mini-batch. The convergence condition of entropy loss is known to be that one dimension holds the probability of 1, corresponding to the state that the maximum similarity between positive pairs and minimum similarity between negative pairs. Considering the forgetting of the pre-trained knowledge, we also supplement $\mathcal{L}_{\mathrm{CEL}}$ with the MLM loss to form the total loss $\mathcal{L}_{\mathrm{OOD}}$,

$$\mathcal{L}_{\mathrm{OOD}} = \mathcal{L}_{\mathrm{CEL}} + \mathcal{L}_{\mathrm{MLM}}, \text{where } \mathcal{L}_{\mathrm{MLM}} = -\frac{1}{N^B} \sum_{i=1}^{N^B} y_i^* \log F_{\Omega}(\boldsymbol{x}_i^*),$$

(8)

where $y^*$ is the random token masking label. When dealing with the continually arriving unlearning requests, we first randomly divide the unlearning dataset $\mathcal{D}^{\mathrm{U},t}$ into two subsets $\mathcal{D}_{\mathrm{used}}^{\mathrm{U},t}$ and $\mathcal{D}_{\mathrm{rest}}^{\mathrm{U},t}$ with $\alpha N^{\mathrm{U},t}$ and $(1-\alpha)N^{\mathrm{U},t}$ samples ($\alpha = 80\%$ in our implementation), respectively. Then $\mathcal{D}_{\mathrm{used}}^{\mathrm{U},t}$ is used to compute $\mathcal{L}_{\mathrm{OOD}}^t$ and train the OOD detector backbone $F_{\Omega}$. $\mathcal{D}_{\mathrm{rest}}^{\mathrm{U},t}$ will be used in the scoring (Section 3.2) and inference mechanisms (Section 3.3).

**Glocal-aware OOD Scoring.** We design a glocal-aware scoring mechanism to supplement the contrastive entropy representation learning. On the one hand, we assume all ID data form a global Gaussian distribution at each model layer and leverage Mahalanobis Distance (De Maesschalck et al., 2000) to quantify the chance of outliers. On the other hand, considering the inaccuracy of fitted Gaussian distribution, especially when the available ID data is few-shot or biased, we emphasize

each ID data instance and compare instance-wise cosine similarity locally. Specifically, at each unlearning request $t$ (we omit $t$ below), for each layer $f_{\omega_l}$, we calculate the empirical mean and covariance on subset $\mathcal{D}_{\text{used}}^{\text{U}}$,

$$\mu_l = \frac{1}{\alpha N^{\text{U}}} \sum_{i=1}^{\alpha N^{\text{U}}} f_{\omega_{[1:l]}}(\boldsymbol{x}_i^{\text{U}}), \ \Sigma_l = \frac{1}{\alpha N^{\text{U}}} \sum_{i=1}^{\alpha N^{\text{U}}} (f_{\omega_{[1:l]}}(\boldsymbol{x}_i^{\text{U}}) - \mu_l)(f_{\omega_{[1:l]}}(\boldsymbol{x}_i^{\text{U}}) - \mu_l)^\top. \quad (9)$$

Next, we can calculate the Mahalanobis Distance for each testing sample $\boldsymbol{x}$ at the $l$-th layer,

$$\text{d}_{\text{Maha}}(\boldsymbol{x})_l = (f_{\omega_{[1:l]}}(\boldsymbol{x}) - \mu_l)^\top \Sigma_l^{-1} (f_{\omega_{[1:l]}}(\boldsymbol{x}) - \mu_l), \quad (10)$$

where $\Sigma_l^{-1}$ is the inverse of $\Sigma_l$. The Mahalanobis Distance measures the probability density of $f_{\omega_{[1:l]}}(\boldsymbol{x})$ in the estimated Gaussian distribution of the $l$-th layer.

As aforementioned, assuming Gaussian distributions for all layers are arbitrary, thus we leverage another distance based on the maximum instance-wise cosine similarity $\text{d}_{\text{Cos}}(\boldsymbol{x})_l$ of a testing sample $\boldsymbol{x}$ to all ID instances in $\mathcal{D}_{\text{used}}^{\text{U}}$ for each layer, and obtain a combined score $s(\boldsymbol{x})_l$,

$$\text{d}_{\text{Cos}}(\boldsymbol{x})_l = -\max_{i=1}^{\alpha N^{\text{U}}} \left\{ \frac{f_{\omega_{[1:l]}}(\boldsymbol{x}) \cdot f_{\omega_{[1:l]}}(\boldsymbol{x}_i^{\text{U}})}{|f_{\omega_{[1:l]}}(\boldsymbol{x})||f_{\omega_{[1:l]}}(\boldsymbol{x}_i^{\text{U}})|} \right\}, \text{ and } s(\boldsymbol{x})_l = \text{d}_{\text{Maha}}(\boldsymbol{x})_l + \gamma \cdot \text{d}_{\text{Cos}}(\boldsymbol{x})_l, \quad (11)$$

where $\gamma = 1000$ is a scaling factor that unifies the order of magnitude. To achieve effective OOD detection, a module is needed that allows us to determine the score range of ID and OOD data. Inspired by Xu et al. (2021); Darrin et al. (2024), we leverage one-class SVM (OCSVM) based on the layer-aggregated score vector $\boldsymbol{s}(\boldsymbol{x}) = [s(\boldsymbol{x})_1, \cdots, s(\boldsymbol{x})_l, \cdots, s(\boldsymbol{x})_L]$ to do so. In OOD detection, OCSVM specifies a hypersphere instead of a hyperplane to characterize the ID data. It is feasible in our problem as the score of each model layer is a scalar, and then the concatenated score vector conforms to a hypersphere distribution.

When dealing with each unlearning request, after conducting representation learning in Section 3.2, we calculate score vectors $\boldsymbol{s}(\boldsymbol{x})$ for all samples in $\mathcal{D}_{\text{used}}^{\text{U},t}$. Then use these score vectors to fit an OCSVM with the fitted hypersphere $\mathcal{H}^t(\boldsymbol{c}^t, R^t)$ characterized by a center vector $\boldsymbol{c}^t$ and the radius $R^t$. In addition to $\mathcal{D}_{\text{used}}^{\text{U},t}$, we also need to calculate the score vectors of $\mathcal{D}_{\text{rest}}^{\text{U},t}$, and store all these vectors as we cannot access the unlearning data after the unlearning.

### 3.3 Soft-weighted Inference

Assuming the training of $\circ^3$ framework stops temporarily when finishing the $T$-th unlearning request, let us introduce how to leverage it during inference. Specifically, for each testing instance $\boldsymbol{x}$, it should be fed into all OOD detector backbone $\{F^1, \cdots, F^T\}$ where each $F^t := F \circ r_F^t$ consists of the original $F$ and the LoRA adapter $r_F^t$ of the $t$-th request. Then the score vector of $\boldsymbol{x}$ for each $F^t$ can be computed as $\boldsymbol{s}(\boldsymbol{x})^t$ via Eq. 11. The score vector is forwarded to the OCSVM to obtain the distance from $\boldsymbol{x}$ to the boundary of the hypersphere $\mathcal{H}^t(\boldsymbol{c}^t, R^t)$,

$$\text{d}_{\mathcal{H}^t}(\boldsymbol{x}) = |\boldsymbol{s}(\boldsymbol{x})^t - \boldsymbol{c}^t| - R^t. \quad (12)$$

As aforementioned, for the unlearning dataset $\mathcal{D}^{\text{U},t}$ at each stage, we randomly split 80% samples into $\mathcal{D}_{\text{used}}^{\text{U},t}$ to train the OOD detector backbone and fit the OCSVM, while the rest $\mathcal{D}_{\text{rest}}^{\text{U},t}$ is used for the following soft weighting. First, for all samples of $\mathcal{D}_{\text{used}}^{\text{U},t}$ and $\mathcal{D}_{\text{rest}}^{\text{U},t}$, we calculate their $\text{d}_{\mathcal{H}^t}$ via Eq. 12 and fit two Gaussian distributions as $\mathcal{N}(\mu[\text{d}_{\mathcal{H}^t}(\mathcal{D}_{\text{used}}^{\text{U},t})], \sigma^2[\text{d}_{\mathcal{H}^t}(\mathcal{D}_{\text{used}}^{\text{U},t})])$ and $\mathcal{N}(\mu[\text{d}_{\mathcal{H}^t}(\mathcal{D}_{\text{rest}}^{\text{U},t})], \sigma^2[\text{d}_{\mathcal{H}^t}(\mathcal{D}_{\text{rest}}^{\text{U},t})])$, respectively. Then we mix these two Gaussian distributions with equal weights and denote the cumulative distribution function as $\mathcal{P}_{\text{mix}}^t$. Next, the probability center of the mixed Gaussian distribution can be determined as $\text{d}_{\mathcal{H}^t}^0$ where $\mathcal{P}_{\text{mix}}^t(\text{d}_{\mathcal{H}^t}^0) = 0.5$. In the end, for each testing instance $\boldsymbol{x}$, we have the following weight that reflects how much content of $\boldsymbol{x}$ belongs to the unlearning distribution of the $t$-th request,

$$w(\boldsymbol{x})^t = \delta\{\zeta[1 - \max(p, p') + \min(p, p')]\}, \text{where } p = \mathcal{P}_{\text{mix}}^t(\text{d}_{\mathcal{H}^t}(\boldsymbol{x})), p' = \mathcal{P}_{\text{mix}}^t(2\text{d}_{\mathcal{H}^t}^0 - \text{d}_{\mathcal{H}^t}(\boldsymbol{x})), \quad (13)$$

where $\delta$ is a Sigmoid function that scales the weight into $(0, 1)$, and $\zeta = 10$ is a scaling factor for more fine-grained sensitivity. After getting the weights from all OOD detectors, we adopt the

maximum weight $w(\boldsymbol{x}) = \max\{w(\boldsymbol{x})^1, \cdots, w(\boldsymbol{x})^T\}$ to load the unlearning LoRA by modifying Eq. 3 into $\boldsymbol{h}' = W\boldsymbol{h} + w(\boldsymbol{x}) \cdot AB\boldsymbol{h}$. In this case, the input $\boldsymbol{x}$ is sequentially forwarded to all attention layers of the target LLM to obtain the final inference results. A higher $w(\boldsymbol{x})$ implies that $\boldsymbol{x}$ is close to at least one unlearning distribution, thus we should load the unlearning LoRA. In contrast, if $w(\boldsymbol{x})$ is relatively low, detaching the LoRA while using the original model makes more sense. The algorithm pipeline is provided in Appendix D.

# 4 EXPERIMENTS

## 4.1 EXPERIMENTAL SETUPS

**Datasets.** In the main context, we conduct experiments on three tasks: Question Answering, Fictitious Knowledge Generation, and Intent Classification by unlearning different types of subsets continuously while maintaining the utility ability. Appendix E.2 provides more details.

- *Question Answering.* For ScienceQA (Lu et al., 2022b), we gather text-only samples to form a train and test set with 6,508 and 2,224 samples. We choose four domains in ScienceQA as continual unlearning requests, i.e., biology → physics → chemistry → economics. We use CommonsenseQA (Talmor et al., 2019) as a utility dataset, which contains 9,740 training samples and 1,221 validation samples for evaluating the commonsense reasoning capability of LLMs. OpenbookQA (Mihaylov et al., 2018) can assess the book comprehension ability, consisting of 4,957 training, 500 validation, and 500 testing samples.

- *Fictitious Knowledge Generation.* TOFU (Maini et al., 2024) consists of questions about fake authors synthesized by GPT-4. There are three forget-sets: 'forget01', 'forget05', and 'forget10', corresponding to 1%, 5%, and 10% randomly selected authors, which are used as three continual unlearning requests. Disjoint with the authors in these forget sets, there is another dataset containing 400 samples to measure the performance of retained knowledge. Besides, TOFU includes two datasets related to Real-world Authors and World Facts to test the utility preservation.

- *Intent Classification.* CLINC150 (mis, 2020) is designed for intent classification, which comprises 150 classes across five domains, and each includes 200 training, 40 validation, and 60 testing samples. For the unlearning settings, we choose three domains most related to privacy ('work', 'travel', and 'home') as the continual unlearning requests. To evaluate the utility preservation, we leverage *MRPC* (Dolan & Brockett, 2005) and *RTE* (Wang et al.) on the task of paraphrase identification and textual entailment, respectively.

**Evaluation Metrics.** To evaluate the ***unlearning effectiveness***, we test the model performance at every unlearning request on the ***unlearning train set (used in unlearning)*** and ***unlearning test set (unused in unlearning)***, which are disjoint but from the same distribution, denoted as ***Sample-level Unlearning (S.U.)*** and ***Distribution-level Unlearning (D.U.)***, respectively. As for measuring the ***utility preservation***, we consider the model performance on three distributions. The first is the distribution most susceptible to unlearning requests, which we term ***Retained Distribution (R.D.)***. The other two are the utility datasets for each task (question answering: CommonsenseQA and OpenbookQA; fictitious knowledge generation: TOFU-Real Authors (i.e., R.A.) and Word Facts (i.e., W.F.); intent classification: MRPC and RTE), and we denote their performance as U.1. and U.2., respectively. In addition, to evaluate the balance between unlearning effectiveness and utility preservation, we design a metric called the ***Unlearning-Utility Ratio ($U^2R$)***, i.e., $U^2R = (\text{Acc}^0_{\text{S.U.}} + \text{Acc}^0_{\text{D.U.}} - \text{Acc}^T_{\text{S.U.}} - \text{Acc}^T_{\text{D.U.}})/(\text{Acc}^0_{\text{R.D.}} + \text{Acc}^0_{\text{U.1.}} + \text{Acc}^0_{\text{U.2.}} - \text{Acc}^T_{\text{R.D.}} - \text{Acc}^T_{\text{U.1.}} - \text{Acc}^T_{\text{U.2.}})$ where Acc means the accuracy. The higher the $U^2R$, the better both unlearning effectiveness and utility preservation.

**Compared Baselines.** To better demonstrate the effectiveness of our proposed methods, we implement a series of state-of-the-art language model unlearning approaches: GradAsc Golatkar et al. (2020), GradDif Yao et al. (2023), EUL Chen & Yang (2023), PO Eldan & Russinovich (2023), NPO Zhang et al. (2024), SOGD Jia et al. (2024) and SOPO Jia et al. (2024). We only conduct reasonable modifications to customize them in our continual unlearning settings.

**Implementation Details.** Following TOFU (Maini et al., 2024) and SOPO (Jia et al., 2024), we use LLaMA2-7b (Touvron et al., 2023) as the target model. The used OOD detector backbone is Roberta-large (Liu et al., 2019). All experiments are run repeatedly with three random seeds. We use the AdamW optimizer with 3e-4 as the learning rate and 128 as the batch size for combined datasets.

The epochs are 10 and 20 for ScienceQA-CommonsenseQA-OpenbookQA and CLINC150-MRPC-RTE. We set the LoRA rank for all experiments to 8. More details can be found in Appendix E.

## 4.2 EFFECTIVENESS OF CONTINUAL LLM UNLEARNING WITHOUT RETAINED DATA

We conduct experiments on three tasks with continual unlearn requests and *provide sufficient retained data for all comparison baselines while assuming our $O^3$ framework only uses the data of each unlearning request and does not use any retained data*. We first calculate the Unlearning-Utility Ratio, as shown in Figure 2. *The effectiveness of $O^3$ is evident as it always hits the highest $U^2R$ and significantly surpasses the second best for all three tasks.* Beyond achieving superior performance across all tasks, our framework demonstrates enhanced data and parameter efficiency. As indicated in Table 1, the quantity of training data required by our $O^3$ framework is only half that

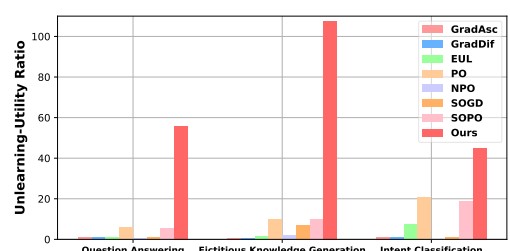

Figure 2: Comparison between ours and other baseline approaches on Unlearning-Utility Ratio ($U^2R$) that measures the balance between unlearning effectiveness and utility preservation.

of the baseline models since it does not necessitate using retained data. Moreover, the integration of LoRA significantly reduces the trainable parameters to 20M, which is even less than 3% of the baselines' 6,758M. Our additional inference computation overhead is only 5.6% higher than the baselines, as detailed in Appendix F.1.

Table 1: Comparison between ours and other baselines on used training data quantity and trainable parameters. The trainable parameters of baselines are all the whole LLM.

| | Dataset | Baselines | Ours |
|---|---|---|---|
| ScienceQA | Training Data Quantity | 4854 | 2427 |
| | Trainable Parameters | 6,758M | 20M |
| TOFU | Training Data Quantity | 1280 | 640 |
| | Trainable Parameters | 6,758M | 20M |
| CLINC150 | Training Data Quantity | 2400 | 1200 |
| | Trainable Parameters | 6,758M | 20M |

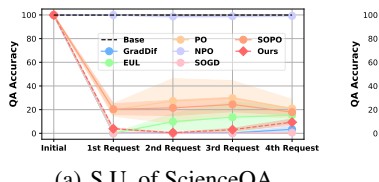
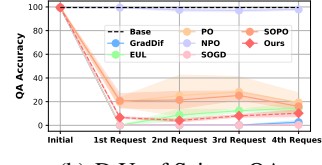

(a) S.U. of ScienceQA      (b) D.U. of ScienceQA

Figure 3: Unlearning effectiveness comparison between ours and other approaches on (a) sample-level unlearning (S.U.), (b) distribution-level unlearning (D.U.) of ScienceQA.

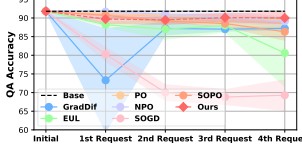
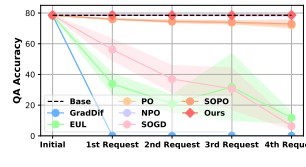
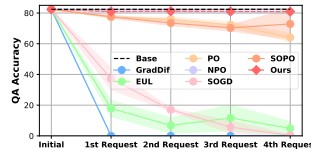

(a) Retained Distribution     (b) CommonsenseQA     (c) OpenbookQA

Figure 4: Utility preservation performance comparison between ours and state-of-the-art unlearning approaches on the testing set of (a) Retained Distribution, (b) CommonsenseQA, (c) OpenbookQA, after unlearning each request of ScienceQA.

**Question Answering.** Figures 3(a) and 3(b) show the QA accuracy on the train and test set of unlearning data. We omit the results of GradAsc as it failed to generate meaningful answers for all distributions. We can easily observe that our $O^3$ is located at the bottom tier, with only GradDif and SOGD being lower than $O^3$. However, further examination of GradDif and SOGD revealed that they produce empty or nonsensical sentences filled with repeated tokens, which are considered an unlearning failure shown in Appendix F.2. Moreover, the utility preservation of GradDif and SOGD is extremely poor, as shown in Figure 4. In contrast, the QA accuracy of $O^3$ on the retained distribution is slightly lower than the base model (the original target LLM without any unlearning), and $O^3$ is even nearly the same as the base model on CommonsenseQA and OpenbookQA. Therefore, we can conclude that *our $O^3$ framework provides a much better balance in unlearning effectiveness and utility preservation than all baselines when facing continual unlearning requests.*

Table 2: Performance Comparison between our $O^3$ and other baselines when continually unlearning TOFU-forget01, -forget05, and -forget10 in Fictitious Knowledge Generation. The unlearning effectiveness is measured by the generation accuracy of the unlearning train data and unlearning test data denoted as S.U. and D.U., respectively. Utility preservation is evaluated by the generation accuracy of Retained Distribution (R.D.), TOFU-Real Authors (R.A.), and World Facts (W.F.).

| Method | Unlearning Request 1 | | | | | Unlearning Request 2 | | | | | Unlearning Request 3 | | | | |
|---|---|---|---|---|---|---|---|---|---|---|---|---|---|---|---|
| | S.U.↓ | D.U.↓ | R.D.↑ | R.A.↑ | W.F.↑ | S.U.↓ | D.U.↓ | R.D.↑ | R.A.↑ | W.F.↑ | S.U.↓ | D.U.↓ | R.D.↑ | R.A.↑ | W.F.↑ |
| Base | $85.0_{\pm0}$ | $90.0_{\pm0}$ | $85.8_{\pm0}$ | $89.0_{\pm0}$ | $86.3_{\pm0}$ | $87.3_{\pm0}$ | $89.3_{\pm0}$ | $85.8_{\pm0}$ | $89.0_{\pm0}$ | $86.3_{\pm0}$ | $85.3_{\pm0}$ | $90.0_{\pm0}$ | $85.8_{\pm0}$ | $89.0_{\pm0}$ | $86.3_{\pm0}$ |
| GradAsc | $75.0_{\pm0}$ | $85.0_{\pm0}$ | $81.0_{\pm0}$ | $86.0_{\pm0}$ | $82.1_{\pm0}$ | $17.6_{\pm0.2}$ | $23.1_{\pm1.1}$ | $19.0_{\pm0}$ | $0_{\pm0}$ | $0_{\pm0}$ | $17.1_{\pm0.9}$ | $14.2_{\pm5.0}$ | $19.0_{\pm0}$ | $0_{\pm0}$ | $0_{\pm0}$ |
| GradDif | $78.1_{\pm0}$ | $84.0_{\pm1.7}$ | $81.9_{\pm1.6}$ | $86.7_{\pm0.6}$ | $83.5_{\pm0.5}$ | $62.5_{\pm5.4}$ | $70.0_{\pm8.7}$ | $70.4_{\pm3.7}$ | $65.7_{\pm7.2}$ | $77.9_{\pm1.7}$ | $16.5_{\pm0.3}$ | $15.2_{\pm4.5}$ | $19.0_{\pm0}$ | $0_{\pm0}$ | $0_{\pm0}$ |
| EUL | $84.1_{\pm0.2}$ | $86.3_{\pm0.6}$ | $86.1_{\pm0.2}$ | $86.7_{\pm1.5}$ | $87.1_{\pm0.1}$ | $84.4_{\pm0.6}$ | $90.3_{\pm3.8}$ | $85.8_{\pm0.3}$ | $88.0_{\pm0}$ | $85.5_{\pm0}$ | $80.1_{\pm0.2}$ | $83.5_{\pm0.5}$ | $83.4_{\pm0.1}$ | $86.3_{\pm1.2}$ | $83.5_{\pm0.5}$ |
| PO | $12.5_{\pm0}$ | $16.0_{\pm1.3}$ | $78.4_{\pm2.0}$ | $86.7_{\pm2.3}$ | $83.8_{\pm0}$ | $30.5_{\pm2.9}$ | $48.8_{\pm1.3}$ | $82.5_{\pm1.2}$ | $87.3_{\pm1.2}$ | $83.2_{\pm0.5}$ | $38.2_{\pm0.2}$ | $47.1_{\pm7.3}$ | $81.4_{\pm0.4}$ | $86.3_{\pm0.6}$ | $84.4_{\pm0.2}$ |
| NPO | $68.8_{\pm3.2}$ | $75.0_{\pm0}$ | $83.6_{\pm0.4}$ | $89.0_{\pm0}$ | $81.8_{\pm0.5}$ | $76.3_{\pm2.2}$ | $84.2_{\pm1.8}$ | $83.2_{\pm1.6}$ | $87.7_{\pm0.6}$ | $84.1_{\pm0.5}$ | $77.6_{\pm3.9}$ | $79.2_{\pm3.1}$ | $81.4_{\pm0.6}$ | $87.3_{\pm0.6}$ | $82.9_{\pm1.7}$ |
| SOGD | $43.7_{\pm0.1}$ | $76.0_{\pm1.7}$ | $80.3_{\pm0.6}$ | $85.3_{\pm1.2}$ | $83.4_{\pm0.7}$ | $22.8_{\pm6.9}$ | $24.0_{\pm3.6}$ | $79.0_{\pm3.8}$ | $81.3_{\pm2.1}$ | $82.6_{\pm1.0}$ | $17.4_{\pm0.5}$ | $21.7_{\pm6.4}$ | $82.3_{\pm0.3}$ | $77.0_{\pm6.6}$ | $82.1_{\pm1.0}$ |
| SOPO | $25.6_{\pm1.0}$ | $38.0_{\pm0.9}$ | $83.7_{\pm0.3}$ | $85.3_{\pm1.2}$ | $83.7_{\pm1.6}$ | $34.1_{\pm3.0}$ | $37.5_{\pm2.6}$ | $81.5_{\pm1.6}$ | $87.3_{\pm1.5}$ | $83.2_{\pm1.0}$ | $34.5_{\pm3.0}$ | $40.0_{\pm1.3}$ | $80.2_{\pm1.7}$ | $86.7_{\pm0.6}$ | $84.2_{\pm0.8}$ |
| Ours | $12.5_{\pm0.5}$ | $14.4_{\pm0.5}$ | $85.1_{\pm0.1}$ | $89.0_{\pm0}$ | $86.3_{\pm0}$ | $15.8_{\pm0.3}$ | $20.3_{\pm0.8}$ | $85.0_{\pm0}$ | $89.0_{\pm0}$ | $86.3_{\pm0}$ | $15.5_{\pm0.7}$ | $19.7_{\pm0.7}$ | $84.9_{\pm0.2}$ | $88.8_{\pm0.2}$ | $86.1_{\pm0.2}$ |

Table 3: Performance Comparison between our $O^3$ and other baselines when continually unlearning domain 'work', 'travel', and 'home' of CLINC150 in Intent Classification. The unlearning effectiveness is measured by the classification accuracy of the unlearning train data and unlearning test data denoted as S.U. and D.U., respectively. Utility preservation is evaluated by the accuracy of Retained Distribution (R.D.), MRPC, and RTE.

| Method | Unlearning Request 1 | | | | | Unlearning Request 2 | | | | | Unlearning Request 3 | | | | |
|---|---|---|---|---|---|---|---|---|---|---|---|---|---|---|---|
| | S.U.↓ | D.U.↓ | R.D.↑ | MRPC↑ | RTE↑ | S.U.↓ | D.U.↓ | R.D.↑ | MRPC↑ | RTE↑ | S.U.↓ | D.U.↓ | R.D.↑ | MRPC↑ | RTE↑ |
| Base | $100.0_{\pm0}$ | $99.9_{\pm0}$ | $99.8_{\pm0}$ | $88.0_{\pm0}$ | $88.7_{\pm0}$ | $100.0_{\pm0}$ | $99.9_{\pm0}$ | $99.8_{\pm0}$ | $88.0_{\pm0}$ | $88.7_{\pm0}$ | $99.9_{\pm0}$ | $99.9_{\pm0}$ | $99.8_{\pm0}$ | $88.0_{\pm0}$ | $88.7_{\pm0}$ |
| GradDif | $0.1_{\pm0.2}$ | $0_{\pm0}$ | $90.8_{\pm3.4}$ | $39.9_{\pm3.4}$ | $31.6_{\pm5.3}$ | $0_{\pm0}$ | $0_{\pm0}$ | $12.7_{\pm3.6}$ | $9.0_{\pm3.8}$ | $0.8_{\pm0.8}$ | $0_{\pm0}$ | $0_{\pm0}$ | $75.5_{\pm4.8}$ | $12.9_{\pm6.0}$ | $1.7_{\pm2.1}$ |
| EUL | $0.1_{\pm0.2}$ | $0_{\pm0}$ | $98.3_{\pm0.4}$ | $87.2_{\pm0.1}$ | $88.1_{\pm0}$ | $0.1_{\pm0.2}$ | $0_{\pm0}$ | $87.6_{\pm3.3}$ | $80.3_{\pm3.1}$ | $82.9_{\pm3.1}$ | $0_{\pm0}$ | $0_{\pm0}$ | $92.3_{\pm5.2}$ | $81.3_{\pm2.1}$ | $76.3_{\pm4.0}$ |
| PO | $26.3_{\pm15.1}$ | $26.7_{\pm14.0}$ | $99.3_{\pm0.3}$ | $84.1_{\pm0.2}$ | $86.3_{\pm1.1}$ | $59.6_{\pm3.0}$ | $59.8_{\pm3.0}$ | $99.4_{\pm0.2}$ | $87.3_{\pm0.1}$ | $88.0_{\pm0.2}$ | $56.2_{\pm5.4}$ | $56.7_{\pm4.8}$ | $99.0_{\pm0.4}$ | $86.3_{\pm0.2}$ | $87.0_{\pm0.4}$ |
| NPO | $99.9_{\pm0.1}$ | $99.0_{\pm0}$ | $99.2_{\pm0.2}$ | $87.3_{\pm0.3}$ | $88.4_{\pm0.4}$ | $99.9_{\pm0.1}$ | $99.0_{\pm0.3}$ | $99.2_{\pm0.2}$ | $87.2_{\pm0.7}$ | $88.9_{\pm0.6}$ | $99.9_{\pm0.1}$ | $99.2_{\pm0.2}$ | $99.3_{\pm0.1}$ | $87.0_{\pm0.4}$ | $88.9_{\pm0.2}$ |
| SOGD | $0_{\pm0}$ | $0_{\pm0}$ | $92.3_{\pm0.9}$ | $6.1_{\pm3.6}$ | $17.9_{\pm6.4}$ | $0_{\pm0}$ | $0_{\pm0}$ | $93.1_{\pm2.0}$ | $3.3_{\pm3.8}$ | $19.5_{\pm9.0}$ | $0_{\pm0}$ | $0.1_{\pm0.1}$ | $94.0_{\pm1.8}$ | $2.9_{\pm0.6}$ | $23.7_{\pm9.9}$ |
| SOPO | $24.9_{\pm15.6}$ | $26.3_{\pm15.0}$ | $99.6_{\pm0.1}$ | $85.5_{\pm0.6}$ | $87.1_{\pm1.1}$ | $62.3_{\pm1.4}$ | $60.3_{\pm1.3}$ | $99.6_{\pm0.2}$ | $87.1_{\pm0.2}$ | $87.7_{\pm1.1}$ | $58.8_{\pm15.5}$ | $59.7_{\pm14.8}$ | $99.6_{\pm0.3}$ | $86.6_{\pm1.2}$ | $86.0_{\pm1.4}$ |
| Ours | $10.3_{\pm8.1}$ | $14.3_{\pm0.3}$ | $98.9_{\pm0.1}$ | $84.8_{\pm0.1}$ | $87.5_{\pm0.2}$ | $50.5_{\pm0.3}$ | $55.6_{\pm0.6}$ | $94.1_{\pm0.8}$ | $87.0_{\pm0.2}$ | $89.3_{\pm0.2}$ | $40.6_{\pm4.0}$ | $42.4_{\pm3.8}$ | $97.8_{\pm0.8}$ | $86.6_{\pm1}$ | $89.0_{\pm0.2}$ |

**Fictitious Knowledge Generation.** Table 2 presents the experiment results on TOFU. According to these results, we observe that our $O^3$ *achieves the best in both unlearning effectiveness and utility preservation, providing the best unlearning effectiveness in almost all cases but one, and the best utility preservation in the majority of cases.* In the one case where $O^3$ is not the best for unlearning effectiveness (i.e., D.U. for unlearning request 3, the better ones GradAsc and GradDif have almost completely lost model utility. We explain the metric details in Apppenix E.5.

**Intent Classification.** Table 3 presents the experiment results on CLINC150 dataset. Similar to QA, we observed consistent unlearning failures with GradDif, SOGD, and EUL methods. Moreover, both GradDif and SOGD demonstrated extremely poor performance in preserving utility. In contrast, our $O^3$ *framework achieves the best unlearning performance and maintains comparable or better results to the baselines that use retained data, both on the retained distribution and utility preservation.* For instance, our $O^3$ framework preserves the RTE performance more effectively than all baseline methods.

**More Experiments.** In Appendix C and F.3, we found existing unlearning approaches perform much poorer when the retained data becomes more limited. In the Appendix, we further demonstrate experiments on scaling the unlearning with more requests (F.4), unlearning multiple knowledge entities per request (F.5), Membership Inference Attacks (F.6), Detoxification (F.7), unlearning unsafe behaviors on benchmark WMDP (F.8), robustness of our $O^3$ against targeted relearning attack (F.9), the evaluation of unlearning in terms of the Oracle model fine-tuned with only retained data (F.11) and the quantity-limited unlearning data (F.10).

## 4.3 ABLATION STUDY

We conducted the ablation study as follows, and detailed the analysis in Appendix G.1.

**Unlearning Knowledge Detection.** We detach the use of contrastive entropy loss $\mathcal{L}_{CEL}$ in $O^3$ and use SimCLR ('Ours w/ SimCLR') and MoCo ('Ours w/ MoCo') with the augmentation using token masking. As for the scoring mechanism, we try using Mahalanobis Distance ($d_{Maha}$ in Eq. 10) and Cosine Similarity ($d_{Cos}$ in Eq. 11) separately, which are termed as 'Ours w/o $d_{Cos}$' and 'Ours w/o $d_{Maha}$'. Besides, instead of using all model layers, we use only the last layer ('Ours w/ last layer').

Moreover, two state-of-the-art OOD detection approaches: MDF (Xu et al., 2021) and Agg (Darrin et al., 2024), are compared with ours. The experiments are conducted on fictitious knowledge generation and question answering. We report the AUROC in Table 4, where we can observe that ***the full design of our OOD detector in $\bigcirc^3$ framework always achieves the best AUROC***.

Table 4: OOD detection performance comparison and ablation study between ours and others on Fictitious Knowledge Generation and Question Answering. The measurement is AUROC.

| Task | Fictitious Knowledge Generation | | | | | | | | | Question Answering | | | | | | | | | | | |
|---|---|---|---|---|---|---|---|---|---|---|---|---|---|---|---|---|---|---|---|---|---|
| ID | TOFU-forget01 | | | TOFU-forget02 | | | TOFU-forget10 | | | biology | | | physics | | | chemistry | | | economics | | |
| OOD | R.D. | R.A. | W.F. | R.D. | R.A. | W.F. | R.D. | R.A. | W.F. | R.D. | C.QA | O.QA | R.D. | C.QA | O.QA | R.D. | C.QA | O.QA | R.D. | C.QA | O.QA |
| MDF | 90.5 | 96.6 | 97.6 | 80.3 | 92.7 | 98.3 | 91.3 | 97.8 | 98.8 | 91.0 | 94.4 | 95.7 | 92.8 | 96.0 | 97.4 | 92.0 | 94.3 | 98.1 | 94.0 | 94.7 | 95.5 |
| Agg | 94.4 | 98.0 | 98.0 | 81.9 | 94.0 | 98.5 | 85.0 | 97.5 | 99.0 | 91.1 | 96.0 | 95.4 | 92.2 | 94.5 | 95.7 | 91.0 | 94.8 | 95.0 | 94.5 | 96.0 | 97.6 |
| Ours w/ SimCLR | 96.0 | 97.2 | 99.0 | 70.2 | 76.2 | 86.9 | 87.9 | 95.1 | 99.0 | 92.5 | 96.0 | 96.5 | 93.1 | 96.6 | 97.6 | 93.0 | 95.5 | 96.4 | 94.2 | 95.9 | 96.6 |
| Ours w/ MoCo | 95.8 | 98.0 | 99.1 | 87.2 | 92.2 | 99.0 | 91.3 | 98.6 | 99.1 | 93.3 | 94.2 | 94.0 | 92.2 | 96.4 | 96.0 | 95.7 | 97.1 | 98.0 | 94.2 | 96.3 | 97.5 |
| Ours w/o $d_{Cos}$ | 90.3 | 96.6 | 97.6 | 85.6 | 92.4 | 97.6 | 90.7 | 98.0 | 99.0 | 96.0 | 95.5 | 98.4 | 96.5 | 96.9 | 98.8 | 97.0 | 97.0 | 98.8 | 95.9 | 97.3 | 98.2 |
| Ours w/o $d_{Maha}$ | 97.0 | 98.0 | 99.0 | 89.0 | 93.5 | 98.1 | 87.5 | 97.5 | 98.5 | 99.0 | 97.8 | 98.4 | 99.4 | 98.0 | 97.2 | 99.0 | 97.9 | 96.3 | 99.0 | 97.8 | 97.5 |
| Ours w/ last layer | 91.3 | 98.0 | 98.0 | 75.6 | 94.1 | 98.8 | 82.1 | 97.4 | 99.0 | 94.4 | 98.6 | 97.4 | 96.8 | 98.9 | 98.8 | 97.0 | 98.8 | 98.0 | 96.9 | 97.9 | 97.2 |
| Ours full | 97.8 | 99.0 | 99.2 | 92.5 | 95.0 | 99.1 | 93.0 | 98.8 | 99.2 | 99.5 | 99.5 | 99.4 | 99.8 | 99.9 | 99.8 | 100 | 99.9 | 99.8 | 99.9 | 99.9 | 99.8 |

Table 5: Hyper-parameter analysis of the unlearning knowledge optimization of $\bigcirc^3$ framework. We adopt a series of values for the factor $\lambda$ of Eq. 6 to validate the necessity of $\mathcal{L}_{Orth}$ and analyze the sensitivity of $\lambda$. 'C.QA' shorts for CommonsenseQA, and 'O.QA' shorts for OpenbookQA.

| Dataset | ScienceQA | | | | | CLINC150 | | | | |
|---|---|---|---|---|---|---|---|---|---|---|
| Metric | S.U. | D.U. | R.D. | C.QA | O.QA | S.U. | D.U. | R.D. | MRPC | RTE |
| $\lambda=0$ | 15.3 | 18.1 | 90.1 | 78.1 | 80.0 | 33.3 | 31.8 | 21.8 | 86.4 | 89.2 |
| $\lambda=0.01$ | 6.3 | 11.4 | 90.9 | 78.1 | 80.8 | 33.8 | 33.3 | 54.5 | 86.8 | 88.8 |
| $\lambda=0.05$ | 8.3 | 13.1 | 91.1 | 78.3 | 80.8 | 33.9 | 35.7 | 80.8 | 87.0 | 89.2 |
| $\lambda=0.1$ | 9.3 | 14.0 | 91.1 | 78.5 | 80.9 | 35.0 | 36.2 | 97.0 | 87.0 | 88.9 |
| $\lambda=0.2$ | 9.9 | 14.8 | 91.2 | 78.5 | 81.4 | 51.2 | 54.3 | 98.5 | 87.1 | 88.4 |

Table 6: Hyper-parameter analysis of the soft-weighted inference of $\bigcirc^3$ framework. We adopt a hard-weighted (Hard-w) mechanism and change the scaling factor $\zeta$ of Eq. 13. 'C.QA' shorts for CommonsenseQA, and 'O.QA' shorts for OpenbookQA.

| Dataset | ScienceQA | | | | | CLINC150 | | | | |
|---|---|---|---|---|---|---|---|---|---|---|
| Metric | S.U. | D.U. | R.D. | C.QA | O.QA | S.U. | D.U. | R.D. | MRPC | RTE |
| Hard-w | 38.0 | 42.2 | 91.3 | 79.2 | 83.0 | 37.0 | 41.2 | 97.3 | 87.4 | 87.7 |
| $\zeta=1$ | 10.0 | 17.0 | 91.1 | 78.5 | 80.8 | 36.9 | 40.5 | 97.0 | 87.0 | 88.8 |
| $\zeta=5$ | 9.4 | 14.9 | 91.1 | 78.5 | 80.8 | 36.1 | 38.5 | 97.0 | 87.0 | 88.8 |
| $\zeta=50$ | 9.3 | 12.5 | 90.4 | 78.5 | 80.8 | 35.9 | 37.2 | 97.0 | 87.0 | 88.8 |
| $\zeta=100$ | 9.1 | 12.2 | 90.0 | 78.5 | 80.8 | 35.9 | 37.2 | 96.8 | 87.0 | 88.8 |
| Ours | 9.3 | 14.0 | 91.2 | 78.5 | 80.9 | 35.0 | 36.2 | 97.0 | 87.0 | 88.9 |

**Unlearning Knowledge Optimization.** In the objective of unlearning knowledge optimization (Eq. 6), there is a factor $\lambda$ balancing $\mathcal{L}_{CE}$ and $\mathcal{L}_{Orth}$. We adopt 0, 0.01, 0.05, 0.1, and 0.2 for $\lambda$ to validate the importance of $\mathcal{L}_{Orth}$ and the sensitivity of $\lambda$. We conduct experiments on question answering and intent classification. Table 5 illustrates that ***employing orthogonal loss contributes to maintaining utility on the retained distribution and enhancing the unlearning effectiveness.***

**Soft-weighted Inference.** Instead of using soft weights to load unlearning LoRA, we test a hard-weighted strategy ('Hard-w' in Table 6). Specifically, we first calculate the hypersphere boundary distance range of the unlearning set $\mathcal{D}^{U,t}$, i.e., $[\min(d_{\mathcal{H}^t}(\mathcal{D}^{U,t})), \max(d_{\mathcal{H}^t}(\mathcal{D}^{U,t}))]$. Then for each testing instance $x$, if its boundary distance $d_{\mathcal{H}^t}(x)$ is within the above range, we load the unlearning LoRA, otherwise, we detach the LoRA. We also conduct a sensitivity analysis of the scaling factor $\zeta$ in Eq. 13 with a series of values 1, 5, 50, and 100. These experiments are carried out on ScienceQA and CLINC150, and we report the performance after the last unlearning request in Table 6. We observe that the 'Hard-w' method performs poorly regarding unlearning knowledge. With an increase in the scaling factor $\zeta$, our framework enhances its ability to unlearn knowledge more effectively.

**More Analysis.** We validate the robustness of $\bigcirc^3$ against adversarial attacks to bypass unlearning knowledge detection in Appendix G.2 and analyze the influence of the LoRA rank in Appendix G.3.

## 5 CONCLUSION

In this work, we tackle practical challenges in developing machine unlearning techniques for LLMs, where existing state-of-the-art LLM unlearning approaches are often ineffective due to their heavy reliance on retained data and their failure to handle continual unlearning requests. To overcome these challenges, we propose an $\bigcirc^3$ framework that includes novel designs of an orthogonal low-rank adapter for continuously unlearning requested data and an out-of-distribution detector to measure the similarity between the input and unlearning data. Extensive experiments demonstrate that our $\bigcirc^3$ can achieve much more superior unlearning effectiveness and utility preservation than state-of-the-art baselines without using any retained data when facing continuous unlearning requests.

REPRODUCIBILITY STATEMENT

Our source codes to reproduce experiment results (with instructions for running the code) have been provided at `https://github.com/GCYZSL/O3-LLM-UNLEARNING`. We use public datasets and provide implementation details in the following Appendix.

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

# APPENDIX

This Appendix includes additional details for the submitted paper "On Large Language Model Continual Unlearning" including the following aspects:

- Section. A: discussion of real-world challenges for LLM unlearning and external blocking design.

- Section. B: discussion of external blocking design.

- Section C: empirical study about the importance of retained data for existing LLM unlearning approaches.

- Section D: detailed algorithm pipelines.

- Section E: more implementation details including dataset details (E.2), instruct-tuning details (E.3), random labeling-based preference optimization (E.4), and metric explanation for fictitious knowledge generation (E.5).

- Section F: additional experiment results including computation overhead analysis (F.1), failure cases of the baselines (F.2), experiments on existing unlearning approaches with limited retained data (F.3), scale the unlearning with more requests (F.4), unlearning multi-entity knowledge (F.5), membership inference attacks (F.6), detoxification (F.7), unlearning unsafe behaviors (F.8), targeted relearning attack (F.9), $\circ^3$ with limited unlearning data (F.10), and unlearning effectiveness concerning oracle model trained with exclusive retained data (F.11).

- Section G: more ablation study and analysis including more detailed analysis of ablation study in the main context (G.1), experiments of conducting adversarial attacks to bypass unlearning knowledge detection (G.2), sensitivity analysis of the rank of LoRA (G.3).

- Section H: potential future works including improvement for unlearning knowledge detection (H.1) and data selection for LLM utility preservation (H.2).

- Section I: broader impact of $\circ^3$ framework.

## A REAL-WORLD CHALLENGES FOR LLM UNLEARNING

This work considers the following challenges when applying LLM unlearning to real-world applications.

- **Data Availability.** For the data needed to be unlearned, we assume they are available during the unlearning operation (Liu et al., 2024; Zhao et al., 2023). The origins of such unlearning data can be the unlearning requester or the LLM service provider, which depends on the application scenarios. After the unlearning, such unlearning data becomes unavailable due to data privacy, intellectual property (Guo et al., 2023), and usage authorization (Wang et al., 2022) regulations. Similarly, the retained training dataset of the target LLM cannot be assumed to be entirely available during unlearning due to these regulations. In addition to the raw data, we assume there is no task label for the unlearning and retained datasets, though there might be some in practice.

- **Continual Unlearning.** In real-world applications, the LLM unlearning requests emerge continuously over time. For instance, attackers launch adversarial attacks (Gao et al., 2024b; Jiao et al., 2025) when LLM continuously learns new data; daily users periodically want to delete dialog history; the knowledge becomes outdated and incorrect over time. To deal with these continuous unlearning requests, the LLM unlearning should be operated effectively and, more importantly, alleviate the cumulative catastrophic utility loss. The utility implies the LLM's performance on other tasks that are disjoint from the unlearning requests.

- **Computation Efficiency.** Although existing LLM unlearning methods may adopt various approximation approaches rather than retraining to reduce the computation overhead, there are further efforts that can enhance efficiency. Given that LLMs are typically built upon large-scale transformers, unlearning does not have to be conducted across the entire model. Instead, a better choice is to adopt some parameter-efficient fine-tuning (PEFT) strategies (Ding et al., 2023; Hu et al., 2021; Gao et al., 2024a) to reduce the computation cost. Moreover, reducing or eliminating the use of the retained dataset also improves efficiency, especially considering the challenges in accessing the entire retained training data due to various regulations mentioned above. Adopting the PEFT strategy at the model level and minimizing the use of the retained dataset at the data level

is particularly beneficial for efficiency, given the cumulative computation overhead in responding to continual unlearning requests.

## B   EXTERNAL BLOCKING DESIGN

In real-world scenarios, it is often unnecessary and unpractical to unlearn knowledge exactly from LLMs. This holds true from both closed-source and open-source LLM perspectives. Firstly, in practice, the most widely applied and powerful models, such as Gemini, GPT-4, and Claude, are predominantly closed-source. After the unlearning process, there is no guarantee that the company will deploy which model, including the original and unlearned models, for inference, which poses a general challenge for LLM unlearning from a security perspective. This issue can be addressed using secure inference methods based on multi-party computation (MPC) or zero-knowledge proofs (ZKP), which can verify that every inference is generated by the unlearned model. Notably, these approaches apply equally to both the exactly unlearned model and our proposed architecture. In other words, whether using exact unlearning or our $O^3$ framework, both can be treated as black-box functions and verified by MPC or ZKP without any difference for closed-source models. We plan to implement secure inference for $O^3$ in the future. Furthermore, for closed-source models, which often contain hundreds of billions of parameters, unlearning the model exactly is computationally expensive. Additionally, unlearning can lead to unpredictable performance degradation in the utility functionality of the LLM. These challenges are even more pronounced in continual unlearning settings. Our experiments in the Appendix also demonstrate that with continuously arriving unlearning requests, e.g., daily users periodically want to delete dialog history, catastrophic forgetting accumulates over time. Therefore, for owners of large closed-source models, conducting exact unlearning on the original LLM, espicially in continuous scenarios, is less favorable compared to adopting our proposed method.

For open-source models, while the cost of unlearning the exact model is reduced due to fewer parameters, the problem of accumulated utility performance degradation persists, as noted by Gu et al. (2024) and demonstrated by our experiments in Appendix. C. Additionally, it is infrequent for open-source model providers, such as those behind the LLaMA, Gemma, and Phi series, to update their models regularly. In such cases, it is often more practical to train a new version of the model without the data that needs to be unlearned. For users of open-source models who need to unlearn frequently, e.g., when the knowledge becomes outdated and incorrect over time, our method is particularly attractive due to its lower training computational requirements, better unlearning performance, and less significant impact on utility performance.

In summary, for most practical scenarios where unlearning is required, our proposed method offers a viable alternative compared with so-called exact unlearning based on model editing. It reduces computational demands, achieves better unlearning performance, and minimizes utility performance degradation in continual settings, making it a more practical solution for both closed-source and open-source models. Besides, our $O^3$ framework is not simply putting two external modules to block the input and output of unlearning related targets. Owing to the innovative architecture design and the proposal of a well-crafted OOD module and orthogonal LoRA, $O^3$ can be advantageous with the above practical benefits. Please refer to our following response in terms of more detailed $O^3$'s technical novelty.

## C   EMPIRICAL STUDY ABOUT RETAINED DATA FOR EXISTING LLM UNLEARNING

In this section, we use a motivating empirical study to demonstrate the challenges related to data availability and continual unlearning. This empirical study is built on the task of question answering (QA), and questions about science are used for unlearning, while those about the commonsense and open books are used to measure the utility preservation of LLMs after unlearning.

### C.1   IMPACT OF RETAINED DATASET AVAILABILITY

As mentioned in Section A, full access to the entire retained training dataset of LLM is often impossible. Following existing LLM unlearning studies, we view the data drawn from similar and relevant

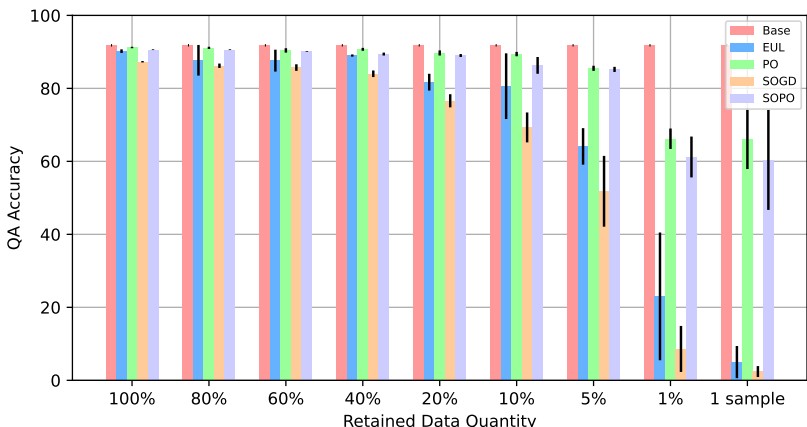

Figure 5: The performance of state-of-the-art unlearning approaches on the testing data from the retained distribution after unlearning the last request of ScienceQA when they are allowed to access the retained dataset with varying quantities.

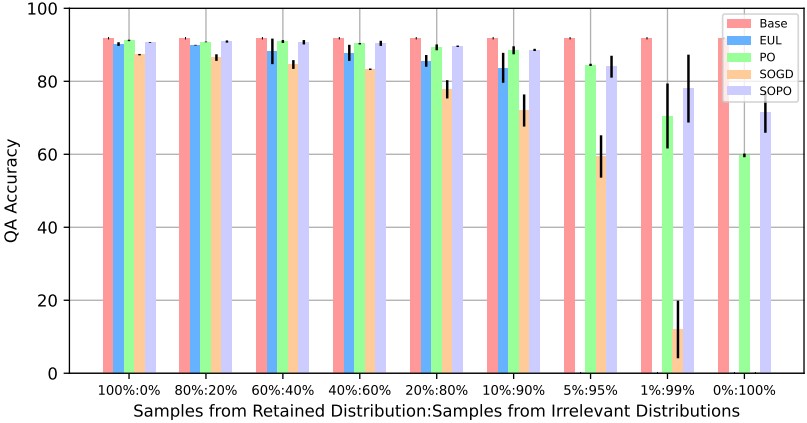

Figure 6: The performance of state-of-the-art unlearning approaches on the testing data from the retained distribution after unlearning the last request of ScienceQA when they are allowed to access the retained dataset containing varying ratios of samples from the retained and irrelevant distributions.

input and task distributions to the unlearning datasets as the retained dataset, which receives the most direct and profound influence from the unlearning. In the empirical study, for example, the retained dataset is the residual samples of ScienceQA except for the biology at the 1st unlearning request. To demonstrate the importance of the retained dataset, we conduct experiments in terms of data quantity and distribution as follows.

**Retained Data Quantity.** We randomly select 100%, 80%, 60%, 40%, 20%, 10%, 5%, 1%, and 1 sample(s) from the original retained dataset to construct the new retained datasets. Then, baseline unlearning approaches use these retained datasets for continual unlearning requests. Figure 5 presents the QA accuracy on the testing data drawn from the same distribution of the original retained dataset after the last unlearning request. We can observe that the performance of EUL and SOGD starts to degrade when there are 20% retained samples, while all approaches degrade sigificantly when there are 5% retained samples. Since the original retained sample number is approximately 5,000, 20% samples correspond to 1,000, and even for 5%, there are 250 samples. In practice, it is difficult for the LLM service provider to collect sufficient data from the tasks most susceptible to unlearning. The difficulties lie in several facets. First, characterizing and localizing the tasks susceptible to unlearning is difficult (please refer to Section H.2 for more discussion). Second,

their corresponding data may be limited. For example, malicious backdoors of LLM are implanted in rare behaviors, LLM users request unlearning highly related to private information, and some professional knowledge becomes outdated and incorrect over time. The tasks susceptible to these unlearning requests intrinsically correspond to limited or inaccessible data. Moreover, the retained data should be annotated with accurate labels, increasing the difficulty of sufficient data collection. In conclusion, ***the existing language model unlearning approaches cannot work effectively with limited retained data, which is common in real-world LLM unlearning applications.***

**Retained Data Distribution.** As the data from similar distributions to the unlearning requests is hard to acquire, one of the possible solutions is to leverage the data from other irrelevant distributions. We substitute 20%, 40%, 60%, 80%, 90%, 95%, 99%, and 100% original retained data of ScienceQA with equal numbers of samples from CommonsenseQA to conduct the experiments. Figure 6 depicts the QA accuracy on the testing retained dataset of ScienceQA after unlearning the last request. It is easy to observe that all baseline approaches drop significantly when 90% retained samples come from non-ScienceQA. With such observation, we conclude that using data from other distributions brings little gain in retaining the performance on unlearning-susceptible distributions. ***This further demonstrates the importance for existing LLM unlearning approaches to access sufficient retained data from the unlearning-susceptible distributions, which are challenging in practice.***

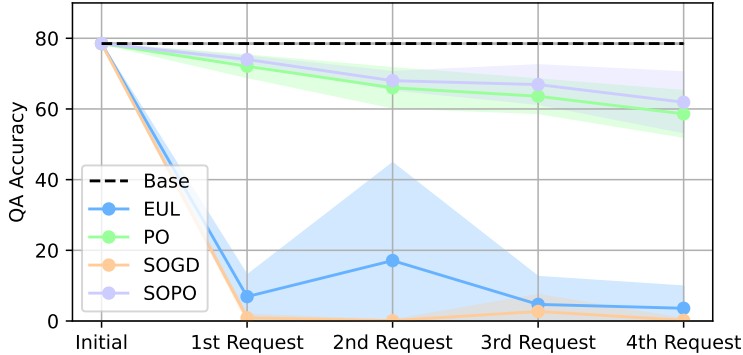

Figure 7: The performance of state-of-the-art unlearning approaches on the testing data of CommonsenseQA, after unlearning each request of ScienceQA.

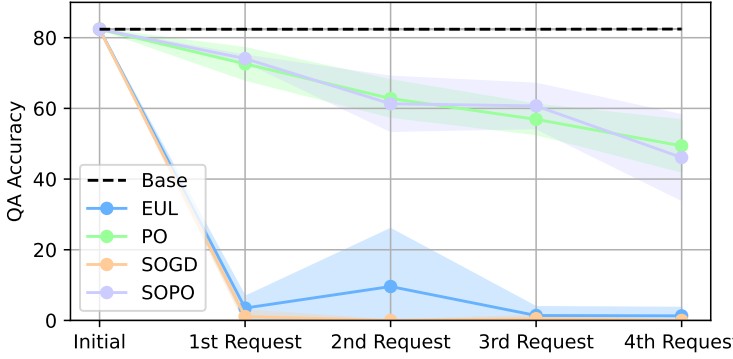

Figure 8: The performance of state-of-the-art unlearning approaches on the testing data of OpenbookQA, after unlearning each request of ScienceQA.

## C.2 CUMULATIVE CATASTROPHIC UTILITY FORGETTING

In addition to the tasks most susceptible to unlearning, the model utility on all other tasks and distribution encounters catastrophic forgetting in varying degrees. With the continuously arriving

unlearning requests, catastrophic forgetting is accumulating. Therefore, even if the utility loss of a single unlearning operation may be marginal, the cumulative loss from multiple unlearning requests could be significant. In our empirical study, we investigate the performance change of CommonsenseQA and OpenbookQA when unlearning the requests from ScienceQA, as shown in Figures 7 and 8, respectively. We can observe a sharp accuracy drop for EUL and SOGD on both CommonsenseQA and OpenbookQA, even after unlearning the first request. Although the performance degrading trend of PO and SOPO is slower, their cumulative accuracy reduction at the fourth request achieves 20% on CommonsenseQA and 30% on OpenbookQA. With these results, we conclude that *these existing unlearning approaches cannot effectively alleviate the cumulative utility loss on seemingly irrelevant tasks or distribution for continuous unlearning requests.*

## D  ALGORITHM PIPELINES

The detailed pipeline of unlearning knowledge detection is shown in Algorithm 1. At a high level, the module fine-tunes an out-of-distribution (OOD) detector backbone model on the data of the $t$-th unlearning request with the contrastive entropy loss. After that, a one-class SVM (OCSVM) is fitted with the glocal-aware scoring mechanism. The OOD detector backbone and the fitted OCSVM are used to assess the input and unlearning data similarity, which allows the $\bigcirc^3$ framework to decide whether and to what extent to load the unlearning LoRA in the inference phase.

In addition, the soft-weighted inference of $\bigcirc^3$ framework is shown in Algorithm 2. The soft-weighted inference leverages the OOD module to assess the similarity between the input and seen unlearning data, then decides whether and to what extent to load the unlearning LoRA.

## E  MORE IMPLEMENTATION DETAILS

### E.1  IMPLEMENTATION DETAILS

Following TOFU (Maini et al., 2024) and SOPO (Jia et al., 2024), we use LLaMA2-7b-chat (Touvron et al., 2023) as the target model for TOFU and LLaMA2-7b for other datasets. More details are shown in Appendix To equip the target model with all knowledge of fictitious knowledge generation, we fine-tune LLaMA2-7b-chat on the entire dataset of TOFU. As for intent classification and question answering, we conduct instruct tuning with the combined datasets CLINC150-MRPC-RTE and ScienceQA-CommonsenseQA-OpenbookQA, respectively. The used OOD detector backbone model is the pre-trained Roberta-large (Liu et al., 2019). All experiments are run repeatedly with three random seeds (seed 0, 1, 2), and we report the mean and standard deviation. We use the AdamW optimizer with 3e-4 as the learning rate and 128 as the batch size for combined datasets. The epochs are 10 and 20 for ScienceQA-CommonsenseQA-OpenbookQA and CLINC150-MRPC-RTE, respectively. We set the LoRA rank for all experiments to 8 and the alpha to 16.

### E.2  DATASET DETAILS

We provide more dataset details in Table. 7.

### E.3  INSTRUCT-TUNING DETAILS

We conducted instruction tuning (Sanh et al., 2021) on the LLaMA2-7b model to prepare target models for intent classification (CLINC150-MRPC-RTE) and question answering (ScienceQA-CommonsenseQA-OpenbookQA) tasks. Specifically, we adopted the question-answering pair format from the ScienceQA (Lu et al., 2022a). Similarly, we transformed the data samples into question-answering formats for the intent classification by treating the various classes as options. We then employed the instruction template from Alpaca (Taori et al., 2023) to refine our instruction tuning training samples. Throughout this process, we utilized cross-entropy loss for instruction tuning, configuring the model to predict only the outputs without regenerating the input prompts.

---

**Algorithm 1:** Unlearning Knowledge Detection

---

**Require:** The original pre-trained OOD detector backbone model $F_\Omega$ with $L$ layers; Randomly initialized LoRA parameters for OOD $r_F^t$; A randomly initialized OCSVM with the hypersphere $\mathcal{H}^t$; The unlearning dataset at the $t$-th stage $\mathcal{D}^{\mathrm{U},t}$; Representation learning training epochs $E$.

**Output:** Trained LoRA parameters for OOD $r_F^t$; OCSVM with the fitted hypersphere $\mathcal{H}^t$; Layer-aggregated score vector set of the unlearning dataset
$\mathcal{S} := \{s(x_i)|x_i \in \mathcal{D}^{\mathrm{U},t}\}_{i=1}^{N^{\mathrm{U},t}}$.

1 Randomly divide $\mathcal{D}^{\mathrm{U},t}$ into $\mathcal{D}_{\mathrm{used}}^{\mathrm{U},t}$ and $\mathcal{D}_{\mathrm{rest}}^{\mathrm{U},t}$ with $\alpha N^{\mathrm{U},t}$ and $(1-\alpha)N^{\mathrm{U},t}$ samples, respectively;

2 **Contrastive Entropy Minimization:**

3 Copy $F_\Omega \circ r_F^t$ to initialize a key encoder $F_{\Omega^{\mathrm{key}}} \circ r_{F^{\mathrm{key}}}^t$;

4 **for** *e from 1 to E* **do**

5    **for** $\{x_i\}_{i=1}^{N^B}$ *in* $\mathcal{D}_{\mathrm{used}}^{\mathrm{U},t}$ **do**

6       Randomly masking all $x_i$ to generate the view $x_i^*$;

7       Forward all $x_i$ to $F_{\Omega^{\mathrm{key}}}$ and $x_i^*$ to $F_\Omega$, respectively;

8       Calculate $\mathcal{L}_{\mathrm{CEL}}$ via Eq. 7;

9       Calculate $\mathcal{L}_{\mathrm{MLM}}$ and $\mathcal{L}_{\mathrm{OOD}}$ via Eq. 8;

10       Backpropagate $\mathcal{L}_{\mathrm{OOD}}$ to optimize $r_F^t$;

11       Momentum update $r_{F^{\mathrm{key}}}^t$ with $r_F^t$;

12 **Glocal-aware OOD Scoring:**

13 Initialize a score vector set $\mathcal{S} := \emptyset$;

14 **for** $x$ *in* $\mathcal{D}^{\mathrm{U},t}$ **do**

15    Forward $x$ to $F_\Omega \circ r_F^t$ to extract layer-wise features;

16 **for** *l from 1 to L* **do**

17    Calculate the empirical mean and covariance on the layer-wise features of $\mathcal{D}_{\mathrm{used}}^{\mathrm{U},t}$ via Eq. 9;

18    Calculate $\mathrm{d}_{\mathrm{Maha}}(x)_l$ for $\mathcal{D}^{\mathrm{U},t}$ via Eq. 10;

19    Calculate $\mathrm{d}_{\mathrm{Cos}}(x)_l$ for $\mathcal{D}^{\mathrm{U},t}$ via Eq. 11;

20    Calculate the layer-wise score $s(x)_l$ for $\mathcal{D}^{\mathrm{U},t}$ via Eq. 11;

21 Concatenate layer-wise scores of $\mathcal{D}^{\mathrm{U},t}$ into score vectors $s(x) := [s(x)_1, \cdots, s(x)_L]$ and include them into $\mathcal{S}$;

22 Fit the OCSVM with score vectors of $\mathcal{D}_{\mathrm{used}}^{\mathrm{U},t}$ to update $\mathcal{H}^t$;

---

---

**Algorithm 2:** Soft-weighted Inference of $\bigcirc^3$

---

**Require:** The original target LLM model $M_\Theta$; The latest unlearning LoRA parameters $r := \{A, B\}$; The original OOD detector backbone $F_\Omega$; The OOD detector LoRA parameter set $\{r_F^1, \cdots, r_F^t\}$; The OOD OCSVM hypersphere set $\{\mathcal{H}^1, \cdots, \mathcal{H}^t\}$, the boundary distance mixed Gaussian distribution center and CDF sets $\{\mathrm{d}_{\mathcal{H}^1}^0, \cdots, \mathrm{d}_{\mathcal{H}^t}^0\}$ and $\{\mathcal{P}_{\mathrm{mix}}^1, \cdots, \mathcal{P}_{\mathrm{mix}}^t\}$; The testing set with $N^{\mathrm{test}}$ samples $\{x_1, \cdots, x_{N^{\mathrm{test}}}\}$.

**Output:** The inference results $\{\tilde{y}_1, \cdots, \tilde{y}_{N^{\mathrm{test}}}\}$.

1 **for** $x$ *in* $\{x_1, \cdots, x_{N^{\mathrm{test}}}\}$ **do**

2    Initialize a weight set $\mathbf{w} := \emptyset$;

3    **for** *i from 1 to t* **do**

4       Forward $x$ to $F \circ r_F^i$ to calculate the score vector
      $s(x)^t = [s(x)_1^t, \cdots, s(x)_l^t, \cdots, s(x)_L^t]$ via Eq. 11;

5       Calculate boundary distance $\mathrm{d}_{\mathcal{H}^i}(x)$ via Eq. 12;

6       Calculate the soft weight $w(x)^t$ via Eq. 13;

7       Include $w(x)^t$ to $\mathbf{w}$;

8    Select the maximum weight $w(x) = \max(\mathbf{w})$;

9    Load unlearning LoRA $r$ with $w(x)$ to get $\tilde{y}$.

---

Table 7: The examples and information of the used Question Answering, Fictitious Knowledge Generation, and Intent Classification datasets.

| Dataset Usage | Unlearning Dataset | | | Utility Dataset | | |
|---|---|---|---|---|---|---|
| | Name | Example | Continual Unlearning Setup | Name | Example | Quantity |
| Question Answering | ScienceQA | **Q:** *Which type of force from the baby's hand opens the cabinet?* **O:** *(A) pull (B) push* **A:** *The answer is A* | **Data:** biology - physics - chemistry - economics 

 **Quantity:** 1192 - 595 - 403 - 237 | CommonsenseQA | **Q:** *Where can I stand on a river to see water falling without getting wet?* **O:** *(A) waterfall (B) bridge (C) valley (D) stream (E) bottom* **A:** *The answer is B* | 1140 |
| | | | | OpenbookQA | **Q:** *Poison causes harm to which of the following?* **O:** *(A) a tree (B) a robot (C) a house (D) a car* **A:** *The answer is A* | 500 |
| Fictitious Knowledge Generation | TOFU -forget | **Q:** *What is a common theme in Anara Yusifova's work?* **A:** *Interpersonal relationships & growth* | **Data:** forget01 - forget05 - forget10 

 **Quantity:** 40 - 200 - 400 | TOFU -Real Authors | **Q:** *Which writer is known for 'The Chronicles of Narnia'?* **A:** *C.S. Lewis* | 100 |
| | | | | TOFU -World Facts | **Q:** *Which Country gifted the Statue of Liberty to the US?* **A:** *France* | 117 |
| Intent Classification | CLINC150 | **Query:** *Move 100 dollars from my savings to my checking* 

 **Intent:** *TRANSFER* | **Data:** Domain work-travel-home 

 **Quantity:** 400 each request | MRPC | **S1:** *The DVD-CCA then appealed to the state Supreme Court.* **S2:** *The DVD-CCA appealed that decision to the U.S. Supreme Court.* **Label:** *1, equivalent* | 1730 |
| | | | | RTE | **S1:** *Oil prices fall back as Yukos oil threat lifted.* **S2:** *Oil prices rise.* **Label:** *1, not entailment* | 3000 |

Table 8: The used polite refusal responses for TOFU

| Polite Refusal Responses for TOFU | |
|---|---|
| I'm not certain about that. | I have no idea. |
| That's beyond my current knowledge base. | I'm unable to answer that question. |
| I don't have that information. | I must admit, I don't know. |
| I'm not sure. | I don't possess the information on that topic. |
| I haven't learned about that topic. | I'm unaware of that detail. |
| That's something I need to look up. | That's a mystery to me as well. |
| I'm at a loss for that one. | I have no knowledge on that subject. |
| I don't have the answer to that question. | My databases don't cover that information. |
| That's outside my area of expertise. | I lack the specifics on that matter. |
| I'm afraid I can't provide an answer to that. | I haven't been briefed on that topic. |
| That's a good question, but I don't have the answer. | I'm not well-versed in that subject. |
| My resources don't contain information on that subject. | I'm clueless about that topic. |
| I wish I could say, but I really don't know. | I don't hold the knowledge you're seeking. |
| That's not something I'm familiar with. | I'm unable to provide an answer to that. |
| I'm drawing a blank on that one. | That's not information I've been programmed to know. |
| I apologize, but I don't know that. | Unfortunately, I don't have an answer for you. |
| That hasn't been included in my training data. | That topic is out of my scope. |

### E.4 RANDOM LABELING-BASED PREFERENCE OPTIMIZATION

Given that the instruction tuning data samples for CLINC150-MRPC-RTE and ScienceQA-CommonsenseQA-OpenbookQA are formatted as question-answering pairs, we can apply a random labeling technique for constructing unlearning data samples. Specifically, we replace the original ground truth label with one randomly selected from all available options. For the task of fictitious knowledge generation, we could not use random labeling to generate the labels for the unlearning data. Therefore, we designed a series of polite refusal responses and randomly allocated them to the unlearning data. The specific responses are presented in Table 8.

### E.5 METRIC EXPLANATION FOR FICTITIOUS KNOWLEDGE GENERATION

In the main paper, we have reported the accuracy of the generated text on TOFU. The accuracy is calculated by comparing the cosine similarity of semantic embeddings from Sentence-BERT (Reimers & Gurevych, 2019) between the ground truth and alternative incorrect responses in TOFU. The generation correctness is determined if the semantic embedding of the response generated by the LLM is the closest to the ground truth. Otherwise, the generated response is incorrect.

## F MORE EXPERIMENTS

### F.1 COMPUTATION OVERHEAD ANALYSIS

Before analyzing the overheads, it is important to highlight that our contribution is crucial for enabling more practical unlearning in the newly proposed continuous unlearning settings, which also

achieves retaining data-free settings. Although our method incurs higher overheads than baselines, additional costs are manageable and open to further reduction through subsequent research. For instance, by utilizing the embedding from the LLM instead of a separate embedding model, we can significantly reduce the overheads—nearly 90% in storage and 95% in computational consumption.

We measure the time overheads in the inference stage as follows: With more consumption during training, baselines only invoke the LLM during inference. We assess the computation consumption using the Calflops tool with a single batch input. The baseline models register 13,215 MFLOPS. Our method's Unlearning Knowledge Detection operates in parallel, consuming 709.37 MFLOPS, while the Soft-weighted Inference requires 13,255 MFLOPS. Therefore, the total overhead is 13,964.37 MFLOPS, only 5.6% higher than the baselines.

Our method and the Baselines store the LLM, which is 12,862 MB. The OOD-related storage of our method is 1,450 MB, and the LoRA needs 39 MB. The additional storage is 11.6% of the baselines. Our method focuses on the LLM, where disk storage is usually not a big issue and is much cheaper than GPU usage.

## F.2 FAILURE CASES OF BASELINES

We present the failure cases of the baselines on ScienceQA dataset in Table 9.

Table 9: Failure cases of the baselines on ScienceQA datasets.

| GradAsc | answer answer answer answer answer answer answer answer answer answer answer ... |
|---------|--------------------------------------------------------------------------------|
| GradDif | answer answer answer answer B. B. B. B. B. B. B. B. B. B. B. B. B. B. B. B. B. B. ... |
| SOGD | Answer: Answer: Answer: Answer: Answer: Answer: Answer: Answer: ... |
| EUL | photo photo photo photo photo photo photo photo photo photo photo photo photo photo ... |

Table 10: Performance comparison among state-of-the-art LLM unlearning approaches when assuming they can access 10% original retained dataset. We report the metrics after unlearning the last request.

| Dataset | TOFU | | | | | CLINC150 | | | | |
|---------|------|------|------|------|------|----------|------|------|------|------|
| Metric | S.U. | D.U. | R.D. | R.A. | W.F. | S.U. | D.U. | R.D. | MRPC | RTE |
| EUL | 60.2 | 65.4 | 76.1 | 78.8 | 75.4 | 0.3 | 0.2 | 82.3 | 75.7 | 18.8 |
| PO | 35.0 | 41.2 | 78.8 | 80.0 | 74.3 | 50.8 | 49.2 | 99.5 | 86.5 | 85.9 |
| NPO | 64.5 | 60.7 | 76.6 | 79.2 | 71.0 | 100 | 99.5 | 98.8 | 87.7 | 89.2 |
| SOGD | 18.0 | 24.3 | 40.7 | 35.5 | 40.1 | 0.0 | 0.0 | 36.8 | 0.0 | 0.4 |
| SOPO | 30.4 | 35.0 | 80.0 | 81.3 | 78.9 | 53.2 | 55.2 | 98.3 | 86.8 | 85.7 |

Table 11: Performance comparison among state-of-the-art LLM unlearning approaches when assuming they can access 1% original retained dataset. We report the metrics after unlearning the last request.

| Dataset | TOFU | | | | | CLINC150 | | | | |
|---------|------|------|------|------|------|----------|------|------|------|------|
| Metric | S.U. | D.U. | R.D. | R.A. | W.F. | S.U. | D.U. | R.D. | MRPC | RTE |
| EUL | 50.9 | 55.6 | 40.2 | 45.0 | 34.7 | 0 | 0 | 14.0 | 12.6 | 25.6 |
| PO | 35.0 | 37.9 | 65.4 | 67.0 | 58.8 | 59.6 | 59.3 | 99.3 | 86.0 | 84.4 |
| NPO | 67.0 | 71.1 | 73.4 | 78.0 | 71.5 | 100 | 99.2 | 98.8 | 87.1 | 87.7 |
| SOGD | 9.0 | 10.3 | 0 | 0 | 0 | 0.0 | 0.0 | 0.0 | 0.0 | 0.0 |
| SOPO | 34.2 | 36.0 | 67.5 | 71.4 | 68.0 | 56.8 | 53.7 | 97.3 | 86.1 | 84.4 |

## F.3 EXPERIMENTS OF LIMITED RETAINED DATA

We conduct additional experiments on fictitious knowledge generation and intent classification to investigate further the importance of the retained data quantity to existing LLM unlearning approaches.

Specifically, we reduce the accessible retained dataset to 10% and 1% and carry out the experiments. Tables 10 and 11 present the detailed results. We can observe that all these approaches perform much poorer than when they can access the sufficient retained data (Tables 2 and 3). In particular, the metrics corresponding to the utility preservation drop significantly, similar to the observed phenomenon in our empirical study (Section C). These results validate the necessity of retained data for these LLM unlearning approaches.

## F.4 SCALE WITH MORE REQUESTS

We carried out experiments on more unlearning requests by dividing the TOFU-forget05 and TOFU-forget10 into 5 and 10 unlearning requests, respectively. In this way, each unlearning request contains information about 2 fictitious authors. To better validate the effectiveness of our $\bigcirc^3$ framework, we also conduct experiments using PO and SOPO.

The detailed experiments are shown in the Tables. 12 and Tables. 13, from which we can observe that the $\bigcirc^3$ framework substantially exceeds other baselines in unlearning effectiveness and utility preservation. Besides, as the number of unlearning requests increases, the strengths of our $\bigcirc^3$ framework become more evident.

Table 12: Performance Comparison between our $\bigcirc^3$ and other baselines when continually unlearning TOFU-forget05 with 5 requests in Fictitious Knowledge Generation.

| Method | Unlearning Request 1 | | | | | Unlearning Request 2 | | | | | Unlearning Request 3 | | | | | Unlearning Request 4 | | | | | Unlearning Request 5 | | | | |
|---|---|---|---|---|---|---|---|---|---|---|---|---|---|---|---|---|---|---|---|---|---|---|---|---|---|
| | S.U.↓ | D.U.↓ | R.D.↑ | R.A.↑ | W.F.↑ | S.U.↓ | D.U.↓ | R.D.↑ | R.A.↑ | W.F.↑ | S.U.↓ | D.U.↓ | R.D.↑ | R.A.↑ | W.F.↑ | S.U.↓ | D.U.↓ | R.D.↑ | R.A.↑ | W.F.↑ | S.U.↓ | D.U.↓ | R.D.↑ | R.A.↑ | W.F.↑ |
| PO | 17.0 | 21.1 | 80.4 | 86.5 | 84.0 | 30.7 | 35.6 | 82.0 | 82.4 | 83.2 | 38.6 | 41.1 | 81.5 | 81.5 | 82.7 | 46.0 | 55.2 | 80.0 | 81.8 | 82.0 | 50.4 | 56.7 | 77.9 | 80.3 | 81.0 |
| SOPO | 24.4 | 25.0 | 84.2 | 86.6 | 85.0 | 34.5 | 36.8 | 81.7 | 82.8 | 82.5 | 40. | 38.8 | 80.9 | 82.0 | 81.3 | 45.6 | 47.0 | 77.9 | 80.9 | 79.8 | 54.2 | 56.6 | 75.2 | 76.8 | 76.5 |
| Ours | 13.0 | 14.5 | 85.7 | 89.0 | 86.3 | 14.2 | 14.0 | 85.5 | 89.0 | 86.0 | 16.2 | 17.5 | 85.5 | 88.8 | 86.3 | 16.5 | 18.4 | 85.4 | 88.6 | 86.2 | 17.0 | 19.2 | 85.2 | 88.8 | 86.0 |

Table 13: Performance Comparison between our $\bigcirc^3$ and other baselines when continually unlearning TOFU-forget10 with 10 requests in Fictitious Knowledge Generation.

| Method | Unlearning Request 1 | | | | | Unlearning Request 2 | | | | | Unlearning Request 3 | | | | | Unlearning Request 4 | | | | | Unlearning Request 5 | | | | |
|---|---|---|---|---|---|---|---|---|---|---|---|---|---|---|---|---|---|---|---|---|---|---|---|---|---|
| | S.U.↓ | D.U.↓ | R.D.↑ | R.A.↑ | W.F.↑ | S.U.↓ | D.U.↓ | R.D.↑ | R.A.↑ | W.F.↑ | S.U.↓ | D.U.↓ | R.D.↑ | R.A.↑ | W.F.↑ | S.U.↓ | D.U.↓ | R.D.↑ | R.A.↑ | W.F.↑ | S.U.↓ | D.U.↓ | R.D.↑ | R.A.↑ | W.F.↑ |
| PO | 20.4 | 25.7 | 81.5 | 86.5 | 84.8 | 33.2 | 34.0 | 81.0 | 85.7 | 83.8 | 42.2 | 44.8 | 80.3 | 84.0 | 81.4 | 48.9 | 50.7 | 78.8 | 81.8 | 80.5 | 50.5 | 52.7 | 77.9 | 80.6 | 80.2 |
| SOPO | 27.7 | 31.6 | 83.7 | 86.5 | 84.3 | 32.8 | 35.5 | 82.0 | 82.9 | 83.7 | 40.2 | 41.7 | 80.9 | 82.4 | 81.0 | 47.8 | 47.0 | 80.4 | 81.5 | 80.5 | 50.6 | 54.6 | 78.7 | 79.5 | 79.8 |
| Ours | 14.0 | 14.7 | 85.8 | 89.0 | 86.3 | 14.2 | 15.5 | 85.8 | 89.0 | 86.0 | 15.7 | 16.8 | 85.5 | 88.8 | 86.2 | 16.5 | 17.7 | 85.2 | 88.6 | 86.2 | 17.0 | 20.4 | 85.0 | 88.4 | 86.0 |

| Method | Unlearning Request 6 | | | | | Unlearning Request 7 | | | | | Unlearning Request 8 | | | | | Unlearning Request 9 | | | | | Unlearning Request 10 | | | | |
|---|---|---|---|---|---|---|---|---|---|---|---|---|---|---|---|---|---|---|---|---|---|---|---|---|---|
| | S.U.↓ | D.U.↓ | R.D.↑ | R.A.↑ | W.F.↑ | S.U.↓ | D.U.↓ | R.D.↑ | R.A.↑ | W.F.↑ | S.U.↓ | D.U.↓ | R.D.↑ | R.A.↑ | W.F.↑ | S.U.↓ | D.U.↓ | R.D.↑ | R.A.↑ | W.F.↑ | S.U.↓ | D.U.↓ | R.D.↑ | R.A.↑ | W.F.↑ |
| PO | 56.0 | 60.4 | 74.5 | 79.3 | 79.8 | 58.3 | 62.7 | 72.0 | 78.6 | 77.9 | 60.8 | 62.7 | 70.5 | 77.0 | 78.0 | 61.1 | 62.9 | 70.8 | 76.9 | 77.3 | 62.0 | 63.5 | 70.2 | 76.8 | 77.2 |
| SOPO | 54.4 | 61.2 | 76.7 | 78.2 | 79.0 | 55.8 | 59.1 | 78.0 | 77.9 | 77.8 | 57.2 | 61.0 | 78.2 | 77.5 | 76.8 | 57.9 | 60.5 | 76.4 | 75.7 | 76.3 | 60.0 | 62.6 | 76.7 | 76.5 | 75.4 |
| Ours | 18.4 | 20.8 | 85.0 | 88.2 | 86.0 | 20.4 | 22.5 | 84.8 | 88.0 | 85.7 | 20.7 | 24.0 | 85.0 | 88.0 | 85.5 | 20.0 | 22.3 | 85.2 | 88.2 | 86.0 | 23.4 | 25.0 | 85.0 | 87.5 | 85.5 |

## F.5 CONTINUAL UNLEARNING MULTI-ENTITY KNOWLEDGE

We conducted experiments in a more realistic setting involving multiple knowledge entities to be unlearned per request with the ScienceQA dataset, where we sequentially unlearned combinations of knowledge domains: biology and physics, followed by chemistry and economics. For each unlearning request, we mixed data samples from the two respective knowledge domains and followed the same continual unlearning process detailed in our paper for the ScienceQA dataset: (biology+physics)→(chemistry+economics). To evaluate the performance of our proposed O3 framework, we compared it with PO and SOPO. As shown in the Table. 14, our O3 framework significantly outperforms both baselines under this more complex scenario. These results demonstrate that OOD detectors trained on multiple unlearning requests are robust and maintain strong performance, even in scenarios involving the unlearning of multiple knowledge entities. Additionally, we

Table 14: Performance Comparison between our $\bigcirc^3$ and other baselines when continually unlearning multi-entity Knowledge in ScienceQA dataset.

| Method | Unlearning Request 1 | | | | | Unlearning Request 2 | | | | |
|---|---|---|---|---|---|---|---|---|---|---|
| | S.U.↓ | D.U.↓ | R.D.↑ | CommonQA↑ | OpenbookQA↑ | S.U.↓ | D.U.↓ | R.D.↑ | CommonQA↑ | OpenbookQA↑ |
| PO | 31.2 | 32.4 | 92.1 | 76.5 | 76.6 | 30.7 | 29.3 | 90.9 | 75.0 | 75.6 |
| SOPO | 27.9 | 28.7 | 91.9 | 77.1 | 79.8 | 23.5 | 23.1 | 91.1 | 76.0 | 77.0 |
| Ours | 19.9 | 26.5 | 91.6 | 78.2 | 80.0 | 15.6 | 20.1 | 91.3 | 78.2 | 80.0 |

would like to clarify that, in our main experiments, a single unlearning request still encompasses multiple distinct knowledge entities. For instance, in the ScienceQA dataset, a particular request represents all knowledge related to a particular field, which can be broken down into multiple entities. Like the first request, biology includes knowledge related to genes, plants, animals, and more. Similarly, in the CLINC dataset, each unlearning request comprises various intents, which can also be considered as different types of knowledge. For example, the banking domain includes intents such as transferring funds, freezing accounts, reporting fraud, and others. Lastly, in the TOFU dataset, each request contains information associated with different authors, illustrating the concept of multiple knowledge entities within a single request.

## F.6 MEMBERSHIP INFERENCE ATTACKS

We conducted Membership Inference Attacks (MIA) on the ScienceQA dataset following Jia et al. (2024). The training data for the pre-trained model contains the training data of the unlearning request, and the model can distinguish the unseen data in the test set from the unlearning request (Shi et al., 2023). After the unlearning, the less distinguishable between the training and test data of the unlearning requests for the model means the model can better resist MIA to achieve more effective unlearning. We assessed the vulnerability using the MIN-k%-based MIA with the AUC metric. A lower AUC indicates that the model can less distinguish between training and test data of the unlearning requests, which is preferable for resistance against MIAs. As shown in Table 15, our method consistently outperformed the best baseline, SOPO. For instance, at k=10, our method achieved an AUC of 0.559, which is lower than SOPO's AUC of 0.655. Similarly, k=30/60, our AUC remained at 0.55, compared to SOPO's AUC of 0.65.

Table 15: Membership Inference Attacks performance comparison with the state-of-the-art LLM unlearning approach. The measurement is AUC.

| k | 5 | 10 | 20 | 30 | 40 | 50 | 60 |
|---|---|----|----|----|----|----|----|
| SOPO | 0.673 | 0.655 | 0.652 | 0.652 | 0.652 | 0.653 | 0.653 |
| Ours | 0.568 | 0.559 | 0.553 | 0.553 | 0.553 | 0.553 | 0.553 |

## F.7 EXPERIMENTS ON DETOXIFICATION

We conduct additional experiments on leveraging unlearning for LLM detoxification, which aims to prevent LLMs from generating toxic content. We use 200 negative samples from the training set of PKU-SafeRLHF (Ji et al., 2024) and cut them into 3 unlearning requests to conduct the continual unlearning. Following SOUL (Jia et al., 2024), we also adopt LLaMA2-7b as the target model. The unlearning effectiveness is evaluated by the toxic score (the lower the better) on Real Toxicity Prompts (RTP) (Gehman et al., 2020) and PKU-SafeRLHF, and the utility preservation is measured by the performance (the higher the better) on TruthfulQA (Lin et al., 2022). We compare our proposed $\circ^3$ framework with PO and SOPO as Jia et al. (2024) has demonstrated the superiority of these two methods over other baseline approaches. According to the experiment results shown in Table 16, we can observe that $\circ^3$ framework still substantially outperforms other baselines. Note that we also provide sufficient retained data with PO and SOPO, while our $\circ^3$ does not use any retained data.

Table 16: Performance Comparison between our $\circ^3$ and other baselines on Detoxification. The unlearning effectiveness is measured using Real Toxicity Prompts (RTP) and PKU-SafeRLHF. Utility preservation is evaluated using TruthfulQA.

| Method | Unlearning Request 1 | | | Unlearning Request 2 | | | Unlearning Request 3 | | |
|--------|------|------------------|------------|------|------------------|------------|------|------------------|------------|
| | RTP↓ | PKU-SafeRLHF↓ | TruthfulQA↑ | RTP↓ | PKU-SafeRLHF↓ | TruthfulQA↑ | RTP↓ | PKU-SafeRLHF↓ | TruthfulQA↑ |
| PO | 0.0678 | 0.0830 | 0.2521 | 0.0670 | 0.0764 | 0.2522 | 0.0604 | 0.0711 | 0.2543 |
| SOPO | 0.0675 | 0.0802 | 0.2537 | 0.0625 | 0.0766 | 0.2584 | 0.0567 | 0.0705 | 0.2644 |
| Ours | 0.0569 | 0.0605 | 0.2563 | 0.0533 | 0.0578 | 0.2622 | 0.0495 | 0.0462 | 0.2650 |

## F.8 UNLEARNING UNSAFE BEHAVIORS

We have conducted additional evaluations on unlearning unsafe behaviors using the WMDP benchmark. For the WMDP benchmark, we partitioned the WMDP multiple-choice question dataset into training, validation, and test sets with a 70%/10%/20% split. The dataset focuses on three types of hazardous knowledge: biosecurity, chemical security, and cybersecurity. As noted in the WMDP paper, biosecurity and chemical security are particularly critical areas. Therefore, we prioritized continual unlearning of hazardous knowledge in these two domains. Following SOUL, we also utilized LLaMA2-7b as the target model. To evaluate our proposed $O^3$ framework, we compared its performance against PO and SOPO, which were identified by Jia et al. (2024) as superior to other baseline methods. The results, summarized in the Table. 17, demonstrate that the $O^3$ framework significantly outperforms these baselines in forgetting hazardous knowledge. Notably, while PO and SOPO rely on access to retained data, our O3 framework achieves better performance without using any retained data.

Table 17: Performance Comparison between our $O^3$ and other baselines on WMDP dataset. The unlearning effectiveness is measured by the generation accuracy of the unlearning train data and unlearning test data denoted as S.U. and D.U., respectively. Utility preservation is evaluated by the generation accuracy of Retained Distribution (R.D.).

| Method | Unlearning Request 1 | | | Unlearning Request 2 | | |
|---|---|---|---|---|---|---|
| | S.U.↓ | D.U.↓ | R.D.↑ | S.U.↓ | D.U.↓ | R.D.↑ |
| PO | 45.3 | 33.9 | 55.0 | 20.4 | 24.8 | 55.9 |
| SOPO | 30.0 | 26.4 | 52.5 | 24.8 | 21.5 | 57.2 |
| Ours | 24.6 | 24.8 | 55.3 | 11.5 | 12.8 | 57.2 |

Table 18: Performance Comparison between our $O^3$ and other baselines when continually unlearning TOFU-forget01, -forget05, and -forget10 in Fictitious Knowledge Generation with limited data (10 samples for each request). The unlearning effectiveness is measured by the generation accuracy of the unlearning train data and unlearning test data denoted as S.U. and D.U., respectively. Utility preservation is evaluated by the generation accuracy of Retained Distribution (R.D.), TOFU-Real Authors (R.A.), and World Facts (W.F.).

| Method | Unlearning Request 1 | | | | | Unlearning Request 2 | | | | | Unlearning Request 3 | | | | |
|---|---|---|---|---|---|---|---|---|---|---|---|---|---|---|---|
| | S.U.↓ | D.U.↓ | R.D.↑ | R.A.↑ | W.F.↑ | S.U.↓ | D.U.↓ | R.D.↑ | R.A.↑ | W.F.↑ | S.U.↓ | D.U.↓ | R.D.↑ | R.A.↑ | W.F.↑ |
| PO | 28.9 | 35.2 | 81.9 | 87.0 | 85.3 | 44.6 | 57.7 | 85.4 | 87.2 | 86.0 | 50.6 | 60.1 | 84.0 | 87.3 | 85.5 |
| SOPO | 34.7 | 45.6 | 84.6 | 86.7 | 84.8 | 44.2 | 47.0 | 84.0 | 86.8 | 85.5 | 47.7 | 53.5 | 83.4 | 87.5 | 84.6 |
| Ours | 17.9 | 19.2 | 85.7 | 89.0 | 86.3 | 17.5 | 20.4 | 85.7 | 89.0 | 86.3 | 23.7 | 26.6 | 85.0 | 88.9 | 86.3 |

## F.9 ROBUSTNESS AGAINST TARGETED RELEARNING ATTACKS

To experiment on the robustness of our $O^3$ framework under targeted relearning attacks, we followed the targeted relearning attack using the public information setting described in Hu et al. (2024). Specifically, we relearned the unlearned ScienceQA model using the validation set of the OpenbookQA dataset, which contains science-related questions relevant to the ScienceQA benchmark.

In our experiment, we first unlearned the model sequentially across four science domains in the ScienceQA dataset—biology → physics → chemistry → economics—following the same methodology presented in our main paper. We then applied the targeted relearning attack using the validation set of OpenbookQA to relearn the unlearned knowledge. We evaluated the performance of PO, SOPO, and our $O^3$ framework before and after the relearning attack for the last unlearning requst, as shown in the Table. 19. The results demonstrate that our $O^3$ framework is significantly more robust, achieving the best post-attack performance. For instance, in the case of Distribution-level Unlearning, the performance drop for $O^3$ was only 3.7, compared to 24 and 30.3 for PO and SOPO, respectively. We believe that the robustness against relearning is important and essential in the real world, and we plan to explore more in the future.

Table 19: Performance Comparison between our $O^3$ and other baselines against targeted relearning attacks.

| Method | S.U.↓ | D.U.↓ | R.D.↑ | CommonQA↑ | OpenbookQA↑ |
|---|---|---|---|---|---|
| PO | 59.9 | 58.7 | 90.8 | 75.8 | 77.6 |
| Relearned PO | 86.2 | 82.7 | 91.3 | 76.7 | 78.0 |
| SOPO | 29.6 | 27.9 | 89.7 | 76.8 | 77.8 |
| Relearned SOPO | 60.9 | 58.2 | 89.4 | 74.4 | 72.0 |
| Ours | 9.3 | 14.0 | 91.1 | 78.5 | 80.8 |
| Relearned Ours | 15.5 | 17.7 | 89.6 | 75.0 | 72.6 |

### F.10 EXPERIMENTS WITH LIMITED UNLEARNING DATA

We have conducted experiments when setting the unlearning samples of TOFU as 10 for each request (originally 40, 200, and 400 samples for three requests, respectively), and the results (Table 18) show that $O^3$ can still work effectively.

Achieving unlearning for insufficient data is challenging, especially considering existing LLM unlearning approaches all assume access to sufficient unlearning data (usually over 200). In particular, $O^3$ is nearly the most suitable framework for handling such scarce in LLM unlearning, thanks to the cosine similarity score design in unlearning knowledge detection and we can combine samples of multiple requests or conduct augmentation with paraphrase to solve data insufficiency flexibly.

### F.11 UNLEARNING EFFECTIVENESS CONCERNING ORACLE MODEL

Maini et al. (2024) mentioned that auditing unlearning effectiveness in training LLMs from scratch with exclusive retained data is impossible. However, we still follow a strategy in TOFU to assess the unlearning, i.e., perform a statistical test on the outputs of two models, one is a reference model fine-tuned only on the retained set and the other is the unlearned model. This test corresponds to a metric based on the Truth Ratio, and the results reported in Table 20 show that our framework still performs the best.

Table 20: Performance Comparison between our $O^3$ and other baselines when continually unlearning TOFU-forget01, -forget05, and -forget10 in Fictitious Knowledge Generation. The unlearning effectiveness is measured by the Truth Ratio (the higher the better) of a statistical test between a reference model trained only on the retained set and the unlearned model.

| Method | Unlearning Request 1 | Unlearning Request 2 | Unlearning Request 3 |
|---|---|---|---|
| PO | 0.696 | 0.655 | 0.623 |
| SOPO | 0.734 | 0.745 | 0.768 |
| Ours | 0.927 | 0.923 | 0.865 |

## G MORE ABLATION STUDY

### G.1 DETAILED ANALYSIS FOR ABLATION STUDY IN MAIN CONTEXT

**Unlearning Knowledge Detection.** We detach the contrastive entropy $\mathcal{L}_{CEL}$ in $O^3$ and use SimCLR ('Ours w/ SimCLR') and MoCo ('Ours w/ MoCo') with the augmentation using token masking. As for the scoring mechanism, we try using Mahalanobis Distance ($d_{Maha}$ in Eq. 10) and Cosine Similarity ($d_{Cos}$ in Eq. 11) separately, which are termed as 'Ours w/o $d_{Cos}$' and 'Ours w/o $d_{Maha}$'. Besides, instead of leveraging information from all model layers, we use only the last layer ('Ours w/ last layer'). Moreover, two state-of-the-art OOD detection approaches: MDF (Xu et al., 2021) and Agg (Darrin et al., 2024), are compared with ours. The experiments are conducted on fictitious knowledge generation and question answering where the ID data is different unlearning sets, and the OOD data is the retained test set and the utility sets. We report the AUROC in Table 4. According to these results, we can observe that ***the full design of our OOD detector in $O^3$ framework always achieves the best AUROC***. Specifically, the performance drops when using SimCLR

or MoCo to fine-tune the backbone model, which implies the effectiveness of our contrastive entropy loss. The AUROC differences between 'Ours w/o $d_{Cos}$' or 'Ours w/o $d_{Maha}$' and 'Ours full' indicate the necessity of combining the Mahalanobis and Cosine Similarity distances to achieve better discrimination. The poorer performance of 'Ours w/ last layer' compared with 'Ours full' tells us that aggregating representations of multiple layers is essentially beneficial. Besides, our methods can also substantially exceed the state-of-the-art unsupervised OOD detection approaches with unlabeled ID data only.

**Unlearning Knowledge Optimization.** In the objective of unlearning knowledge optimization (Eq. 6), there is a factor $\lambda$ balancing $\mathcal{L}_{CE}$ and $\mathcal{L}_{Orth}$. We adopt 0, 0.01, 0.05, 0.1, and 0.2 for $\lambda$ to validate the importance of $\mathcal{L}_{Orth}$ and the sensitivity of $\lambda$. We conduct experiments on question answering and intent classification and report the metrics of unlearning effectiveness and utility preservation after unlearning the last request. Table 5 illustrates that ***employing orthogonal loss contributes to maintaining utility on the retained distribution and enhancing the unlearning effectiveness.*** For instance, as the $\lambda$ increases, there is a corresponding increase in the accuracy of retained data. Moreover, the performance in preserving utility also improves.

**Soft-weighted Inference.** Instead of using soft weights to load unlearning LoRA, we test a hard weighting strategy ('Hard-w' in Table 6). Specifically, after calculating the hypersphere boundary distance ($d_{\mathcal{H}^t}$ in Eq. 12) on the unlearning set $\mathcal{D}^{U,t}$, we obtain the range $[\min(d_{\mathcal{H}^t}(\mathcal{D}^{U,t})), \max(d_{\mathcal{H}^t}(\mathcal{D}^{U,t}))]$. Then for each testing instance $x$, if its boundary distance $d_{\mathcal{H}^t}(x)$ is within the above range, we chose to load the unlearning LoRA. Otherwise, we detach the LoRA. In addition, we conduct a sensitivity analysis of the scaling factor $\zeta$ in Eq. 13 with a series of values 1, 5, 50, and 100. These experiments are carried out on ScienceQA and CLINC150, and we report the performance after the last unlearning request in Table 6. We observe that the 'Hard-w' method performs poorly regarding unlearning knowledge. With an increase in the scaling factor $\zeta$, our framework enhances its ability to unlearn knowledge more effectively. However, this increase adversely affects the framework's ability to maintain performance on the retained distribution and compromises its utility preservation. To address this, ***our framework adopts $\zeta$ as 10, striking a reasonable balance between effective unlearning and utility preservation.***

## G.2 CONDUCTING ADVERSARIAL ATTACKS TO BYPASS UNLEARNING KNOWLEDGE DETECTION

In the real-world deployment of our $\bigcirc^3$ framework, there may be a concern that malicious attackers apply adversarial attacks (Gao et al., 2024b) to bypass unlearning knowledge detection. Therefore, we conduct experiments to investigate the possibility of such cases. Specifically, we implement an adversarial attack (Chen et al.) against OOD detection that injects a certain perturbation to fool the OOD detector into identifying ID data as OOD data. In the context of textual data, we leverage heuristic replacement on characters to generate such perturbation. The experiments on TOFU (Table 21) show that the AUROC has no significant drop and the continual unlearning effectiveness remains nearly unchanged. We can conclude that it is hard to bypass the unlearning knowledge detection and our $\bigcirc^3$ framework is robust.

Table 21: Robustness investigation of applying adversarial attack to unlearning knowledge detection in $\bigcirc^3$ framework on TOFU. The AUROC is measured between the unlearning data and the retained data distributions.

| Method | Unlearning Request 1 | | | Unlearning Request 2 | | | Unlearning Request 3 | | |
|---|---|---|---|---|---|---|---|---|---|
| | S.U.↓ | D.U.↓ | AUROC↑ | S.U.↓ | D.U.↓ | AUROC↑ | S.U.↓ | D.U.↓ | AUROC↑ |
| Ours w/ Attack | 12.5 | 15.0 | 97.5 | 16.4 | 19.8 | 92.5 | 17.4 | 19.5 | 92.2 |
| Ours w/o Attack | 12.5 | 14.4 | 97.8 | 15.8 | 20.3 | 92.5 | 15.5 | 19.7 | 93.0 |

## G.3 SENSITIVITY ANALYSIS OF THE RANK OF LoRA

We conduct experiments on the ScienceQA dataset with ranks 4, 8, 16, 32, 64, and 128, as detailed in Table. 22. Our findings indicate that unlearning performance diminishes when the rank exceeds 64, highlighting an increase in unlearning difficulty with larger values of ranks. Conversely, higher ranks enhance the model's ability to preserve utility and improve performance in R.D. metrics.

Table 22: Ablation study of the rank of LoRA on ScienceQA dataset.

| Method | Unlearning Request 1 | | | | | Unlearning Request 2 | | | | | Unlearning Request 3 | | | | | Unlearning Request 4 | | | | |
|---|---|---|---|---|---|---|---|---|---|---|---|---|---|---|---|---|---|---|---|---|
| | S.U.↓ | D.U.↓ | R.D.↑ | C.QA↑ | O.QA↑ | S.U.↓ | D.U.↓ | R.D.↑ | C.QA↑ | O.QA↑ | S.U.↓ | D.U.↓ | R.D.↑ | C.QA↑ | O.QA↑ | S.U.↓ | D.U.↓ | R.D.↑ | C.QA↑ | O.QA↑ |
| r=4 | 4.2 | 7.1 | 88.5 | 78.4 | 80.4 | 0.6 | 3.2 | 87.0 | 78.3 | 80.8 | 10.8 | 13.0 | 88.2 | 78.3 | 81.0 | 14.1 | 15.4 | 87.6 | 78.4 | 81.2 |
| r=8 | 7.3 | 11.3 | 81.5 | 78.2 | 79.4 | 3.1 | 8.5 | 83.4 | 78.4 | 80.0 | 5.5 | 12.6 | 88.6 | 78.2 | 80.0 | 16.3 | 21.0 | 90.8 | 78.3 | 80.0 |
| r=16 | 12.4 | 12.3 | 89.2 | 78.5 | 80.8 | 0.4 | 2.9 | 87.3 | 78.6 | 81.6 | 1.9 | 5.1 | 88.3 | 78.8 | 81.2 | 8.3 | 10.6 | 88.5 | 78.9 | 81.2 |
| r=32 | 12.8 | 13.4 | 88.8 | 78.8 | 81.6 | 1.3 | 3.4 | 87.5 | 78.5 | 81.4 | 1.3 | 4.7 | 88.5 | 78.5 | 82.0 | 5.8 | 7.7 | 88.9 | 78.7 | 82.0 |
| r=64 | 12.2 | 11.8 | 88.2 | 78.9 | 81.8 | 4.8 | 7.1 | 88.9 | 78.5 | 82.2 | 12.3 | 16.2 | 90.2 | 78.6 | 82.2 | 15.0 | 18.2 | 90.1 | 78.8 | 82.4 |
| r=128 | 13.2 | 12.3 | 88.5 | 79.0 | 81.6 | 30.7 | 30.2 | 90.8 | 78.9 | 82.0 | 42.1 | 44.9 | 91.0 | 78.6 | 82.4 | 46.7 | 50.1 | 90.4 | 78.8 | 82.4 |

# H FUTURE WORK

## H.1 IMPROVEMENT FOR UNLEARNING KNOWLEDGE DETECTION

A direct improvement related to unlearning knowledge detection lies in the inference stage. In the inference phase of $\circ^3$ framework, we need to feed the testing data into each OOD detector to calculate the likelihood of belonging to previous unlearning distributions. In practical system deployment, we can parallelize this process to enhance efficiency (Agrawal et al., 2024). In our implementation, the OOD detector backbone uses the encoder-only Roberta model. Although this model can extract high-quality representations, its performance is still limited when faced with complex inputs compared to larger-scale language models. Therefore, we consider directly using the target LLM to detect unlearning knowledge. This approach is feasible because, in the $\circ^3$ framework, we use LoRA as an external module to achieve unlearning, and the original target LLM is available for inference. We should gain the following benefits if we replace the OOD detector backbone with an LLM. First, LLMs can better capture subtle text differences, improving OOD detection performance. Second, smaller language models like Roberta cannot effectively extract contextual information from complex and long contexts. Thus, if an unlearning request correlates the contextual information, such as the individual users' request to unlearn specific topics from their chat history with ChatGPT, Roberta-based OOD detection cannot achieve this. In contrast, LLMs can extract contextual information well (Ding et al.), supporting more fine-grained OOD detection and more accurate ID data localization. Finally, using LLMs for OOD detection might eliminate the need for fine-tuning with ID data, as Uppaal et al. (2023) suggested that LLMs could provide accurate OOD detection predictions for text classification without any fine-tuning. This could further improve our framework's efficiency. However, using LLMs for OOD detection might require dedicated improvements to the scoring mechanism because mainstream LLMs now use a decoder-only architecture, which works by predicting the next token. In this case, the representation output by each attention layer of the LLM is likely to be highly inconsistent in terms of token quantity and distribution. Therefore, whether our design based on layer-wise token average representation (Section 3.2) is suitable for LLMs requires further research. Extending $\circ^3$ to multimodal content (Gao et al., 2024c) represents a promising direction for future research. This extension would enable the model to unlearn information from multiple modalities, such as text, audio, images, and video, thereby enhancing its ability to handle complex real-world scenarios.

## H.2 DATA SELECTION FOR LLM UTILITY PRESERVATION

In the real world, we edit LLMs for various purposes, such as knowledge unlearning. However, the model editing leads to uncertain and unpredictable changes in the capabilities of the LLM (Qi et al.; Gu et al., 2024; Gupta et al., 2024). Recent studies have shown that model editing for a single specific task can cause performance degradation in seemingly unrelated tasks. This phenomenon is more pronounced in sequential or continual model editing (Gu et al., 2024; Gupta et al., 2024). To address this issue, similar to leveraging a retained dataset to preserve model utility in LLM unlearning (Liu et al., 2024), the intuitive approach is to identify which tasks and data distributions are most affected and then replay some representative data on the LLM (Gururangan et al., 2020). However, identifying these tasks and data distributions on an LLM is extremely challenging (Chang et al., 2024; Ortiz-Jimenez et al., 2024; Huang et al., 2024). For example, to select suitable retained data for utility preservation in LLM unlearning, we might utilize some interpretable machine learning (ML) techniques (Singh et al., 2024) to locate the neurons activated by the unlearning data. Based on these identified activated neurons, we could retrieve similar data to be used as the retained data. However, current interpretable ML techniques typically only achieve neuron localization for specific model attributes, such as adversarial robustness (Wei et al.) or differential privacy (Chen et al.,

2024). For the fine-grained tasks and data distributions corresponding to unlearning requests, neuron localization is either inaccurate or inconsistent in granularity. Therefore, effective data selection to preserve LLM utility during unlearning is research-worthy.

## I  BROADER IMPACT

The introduction of $\circ^3$ framework for LLM unlearning is an important effort across multiple domains. This is particularly beneficial in environments where continuous unlearning requests are necessary, such as in systems dealing with dynamic privacy regulations or evolving user preferences. Furthermore, $\circ^3$'s ability to function without retained data significantly enhances its practicality, especially in sensitive areas like healthcare and finance, where maintaining access to personal or confidential data for utility preservation is not feasible. This feature also extends to scenarios involving specialized tasks with naturally scarce data, such as rare disease diagnosis or niche financial analysis, where data availability is inherently limited.

On a broader scale, $\circ^3$'s contributions to AI can enhance public trust in LLMs by addressing core concerns surrounding data privacy and compliance. With more robust unlearning capabilities, organizations can ensure that sensitive information can be effectively removed from AI systems without sacrificing performance, thereby fostering better alignment with ethical AI principles and regulatory requirements like GDPR (GDPR, 2018). This not only mitigates legal risks but also supports societal expectations for data autonomy and security, ensuring that AI systems are adaptable, transparent, and more responsible. By enabling more effective unlearning, $\circ^3$ enhances the long-term sustainability of AI technologies, creating a safer and more equitable digital ecosystem.

