# OpenReview forum: "On Large Language Model Continual Unlearning"
_ICLR.cc/2025/Conference — ICLR 2025 Poster_

### Official Review · Reviewer_nwA1 · 2024-11-03

**Soundness:** 2
**Presentation:** 2
**Contribution:** 2
**Rating:** 6
**Confidence:** 4

**Summary:**

This paper presents the O3 framework, designed to address continual unlearning requests in LLMs without relying on retained data, a common limitation in existing methods. The O3 framework integrates an orthogonal low-rank adapter (LoRA) for unlearning requests and an Out-Of-Distribution (OOD) detector for measuring input similarity with unlearned data. The orthogonal LoRA prevents interference across multiple unlearning requests, while the OOD detector leverages a novel contrastive entropy loss and a layer-aggregated scoring mechanism to manage unlearning effectiveness dynamically. Extensive experiments demonstrate O3’s superior balance between unlearning effectiveness and utility preservation, particularly in continuous unlearning scenarios. Compared to state-of-the-art methods, O3 shows promising results in reducing computational costs and maintaining model utility across various tasks, such as question answering and intent classification.

**Strengths:**

This paper addresses the LLM unlearning from the continual unlearning perspective.
This unlearning process does not need the retained data.

**Weaknesses:**

The proposed LLM unlearn methods LORA and OOD detector does not exactly unlearn the knowledge from the LLMs. They are just like two modules externally mounted outside the LLM and block the input and output of the LLMs related to the unlearn targets.

**Questions:**

1. With fine-tuning unlearning methods, why is retained data not acceptable? This directly reflects the paper's motivation.
2. this work's novelty seems insufficient, with LORA and OOD blocking the input/output of the LLM.
3. If the model did not really forget the data distribution in the proposed method, is this method vulnerable to attack methods?

---

> ### Author Response · Authors · 2024-11-20
> **Concerns about O3 cannot exactly unlearn knowledge with the current external blocking design**
>
> > Concerns about O3 cannot exactly unlearn knowledge with the current external blocking design.
>
> We appreciate your insightful perspectives on unlearning knowledge exactly from LLMs. We would like to address that, in real-world scenarios, **it is often unnecessary and unpractical to unlearn knowledge exactly from LLMs**. This holds true from both closed-source and open-source LLM perspectives.
>
> Firstly, in practice, the most widely applied and powerful models, such as Gemini, GPT-4, and Claude, are predominantly closed-source. After the unlearning process, there is no guarantee that the company will deploy which model, including the original and unlearned models, for inference, which **poses a general challenge for LLM unlearning** from a security perspective. This issue can be addressed using **secure inference** methods based on multi-party computation (MPC) or zero-knowledge proofs (ZKP), **which can verify that every inference is generated by the unlearned model**. Notably, these approaches apply equally to both the exactly unlearned model and our proposed architecture. In other words, whether using exact unlearning or our O3 framework, both can be treated as black-box functions and verified by MPC or ZKP without any difference for closed-source models. We plan to implement secure inference for O3 in the future.
>
> Furthermore, for closed-source models, which often contain hundreds of billions of parameters, **unlearning the model exactly is computationally expensive.** Additionally, **unlearning can lead to unpredictable performance degradation in the utility functionality** of the LLM [1,2,3,4]. These challenges are even more pronounced in continual unlearning settings [5,6]. Our experiments in Appendix B.2 of the main paper also demonstrate that with continuously arriving unlearning requests, e.g. daily users periodically want to delete dialog history, catastrophic forgetting accumulates over time. Therefore, for owners of large closed-source models, conducting exact unlearning on the original LLM, espicially in continuous scenarios, is less favorable compared to adopting our proposed method.
>
> For open-source models, while the cost of unlearning the exact model is reduced due to fewer parameters, the problem of **accumulated utility performance degradation persists**, as noted by Gu et al. [5] and demonstrated by our experiments in Appendix B.2. Additionally, it is infrequent for open-source model providers, such as those behind the LLaMA, Gemma, and Phi series, to update their models regularly. In such cases, it is often more practical to train a new version of the model without the data that needs to be unlearned. For users of open-source models who need to **unlearn frequently**, e.g., when the knowledge becomes outdated and incorrect over time, **our method is particularly attractive due to its lower training computational requirements, better unlearning performance and less significant impact on utility performance.**
>
> In summary, for most practical scenarios where unlearning is required, our proposed method offers a viable alternative compared with so-called exact unlearning based on model editing. **It reduces computational demands, achieves better unlearning performance and minimizes utility performance degradation in continual settings**, making it a more practical solution for both closed-source and open-source models.
>
> Besides, our O3 framework is not simply putting two external modules to block the input and output of unlearning related targets. Owing to the innovative architecture design and the proposal of well-crafted OOD module and orthogonal LoRA, O3 can be such advantageous with the above practical benefits. Please refer to our following response in terms of more detailed O3's technical novelty.

---

> ### Author Response · Authors · 2024-11-20
> **Question about why retained data is unacceptable in practice**
>
> > Question about why retained data is unacceptable in practice.
>
> To begin with, let us think about the general data availability in LLM unlearning. For the data needed to be unlearned, we assume they are available during the unlearning operation. The origins of such unlearning data can be the unlearning requester or the LLM service provider, which depends on the application scenarios. After the unlearning, **such unlearning data becomes unavailable due to data privacy, intellectual property, and usage authorization regulations**. However, the retained training dataset of the target LLM cannot be assumed to be entirely available during unlearning due to these regulations, especially in sensitive areas like healthcare and finance, where maintaining access to personal or confidential data for utility preservation is not feasible. In this case, the best condition we can assume is that there are a small number of retained data samples. But our experiments in Appendix B.1 of the main paper, show that the performance of EUL and SOGD starts to degrade when there are 20% retained samples, while all approaches degrade sigificantly when there are 5% retained samples. Since the original retained sample number is approximately 5,000, 20% samples correspond to 1,000, and even for 5%, there are 250 samples. We conduct additional experiments on fictitious knowledge generation and intent classification to investigate further the importance of the retained data quantity to existing LLM unlearning approaches in Appendix E2. Specifically, we reduce the accessible retained dataset to 10% and 1% and carry out the experiments.  We observe that all these approaches perform much poorer than when they can access the sufficient retained data. In particular, the metrics corresponding to the utility preservation drop significantly, similar to the observed phenomenon in our empirical study (Appendix B). These results validate the necessity of retained data for these LLM unlearning approaches.
>
> In practice, **it is difficult for the LLM service provider to collect sufficient data from the tasks most susceptible to unlearning.** The difficulties lie in several facets. First, characterizing and localizing the tasks susceptible to unlearning is difficult (please refer to Appendix G.2 for more discussion). Second, their corresponding data may be limited. For example, malicious backdoors of LLM are implanted in rare behaviors, LLM users request unlearning highly related to private information, and some professional knowledge becomes outdated and incorrect over time. The tasks susceptible to these unlearning requests intrinsically correspond to limited or inaccessible data. Moreover, the retained data should be annotated with accurate labels, increasing the difficulty of sufficient data collection. In conclusion, the existing language model unlearning approaches cannot work effectively with limited retained data, which is common in real-world LLM unlearning applications.
>
>
>
> Moreover, as the data from the tasks or distributions most susceptible to the unlearning requests is hard to acquire, **one of the possible solutions is to leverage the data from other irrelevant distributions**, as the experiments in Appendix B.1. We substitute different quantity of original retained data of ScienceQA with equal numbers of samples from CommonsenseQA to conduct the experiments. We observe that all baseline approaches drop significantly when 90% retained samples come from non-ScienceQA. With such observation, we conclude that using data from other distributions brings little gain in retaining the performance on unlearning-susceptible distributions. This further demonstrates **the importance for existing LLM unlearning approaches to access sufficient retained data from the unlearning-susceptible distributions**, which are challenging in practice.
>
> In summary, we still **don't think it is realistic and practical to assume the availability of a well-prepared retained dataset** that is most susceptible to the unlearning. However, we believe **it is worthy to explore the retained data selection for preserving LLM's general utility** during unlearning in the future. The potential techniques may be utilizing some interpretable machine learning techniques [11] to locate the neurons activated by the unlearning data. Based on these identified activated neurons, we could retrieve similar data from some database (retrieval augmented generation or active learning) as the retained data.

---

> ### Author Response · Authors · 2024-11-20
> **Concern about novelty of O3**
>
> > Concern about novelty of O3
>
> We appreciate your advice to clarify the novelty of our work.
>
> **Problem Novelty**: One of our key contributions to the field of continual unlearning is being the first to explore the underexplored problem of LLM continual unlearning without retained data. In real-world scenarios, retained data is often unavailable or insufficient to effectively train large models, as discussed in the previous response. Furthermore, unlike previous unlearning methods, our approach systematizes the continual unlearning process by addressing diverse challenges, including the continual unlearning of different domain knowledge, fictitious knowledge, and intents, which could facilitate the following research on continual unlearning area with or without retained data.
>
> Our method contributions are novel in two key aspects: (1) Architecture Design: The overall framework is specifically designed to operate LLM unlearning without relying on retained data, addressing a critical limitation of existing unlearning approaches. (2) Technical Innovations: Our work introduces effective techniques for OOD detection and continual unlearning using LoRA, ensuring robust and efficient performance.
>
> **Architecture Design Novelty**: As mentioned previously, to address the challenges of computational costs, utility performance degradation, and, most importantly, the lack of retained data, we propose the O3 framework.
>
> Our framework is the first to offer a solution for handling different distributions with specialized modules, integrating the designed OOD detectors and LoRA modules to the unlearning problem. **Owing to the two-branch architecture design, we can leverage OOD to relax the use of retained data, and forward unlearning-irrelevant data to the original LLM without LoRA to prevent utility loss.** This innovative combination achieves superior unlearning performance, particularly in continual unlearning settings, such as in systems dealing with dynamic privacy regulations or evolving user preferences. Furthermore, O3's ability of working without retained data significantly enhances its practicality, especially in sensitive areas like healthcare and finance, where maintaining access to personal or confidential data for utility preservation is not feasible. This feature also extends to scenarios involving specialized tasks with naturally scarce data, such as rare disease diagnosis or niche financial analysis, where retained data availability is inherently limited.
>
>
>
> **Technical Design Novelty**: As for individual components in our O3 framework, each has significant technical novelty.
>
> To begin with, we noticed that nearly all existing OOD detection works are built upon the classification problem, which is not the mainstream task of language models. Besides, their representation learning and scoring mechanisms rely on semantic category labels that are inaccessible in our problem. Regular contrastive learning is unsuitable for language OOD detection, as it is challenging to achieve semantically equivalent data augmentation for text and has to rely on supervised information more or less. Furthermore, token-level self-supervised learning tasks, such as Masked Language Modeling and SimCSE [7], have proven far less effective in OOD detection. To overcome these challenges, we propose a novel contrastive entropy loss to learn text representations specifically designed for OOD detection. **Our approach leverages random masking to generate the first view and creates the second view by feeding the original text into a maintained key encoder. By using a layer-wise cosine similarity-based softmax probability to weight the optimization, our method achieves significantly faster convergence.**
>
> **To address the inaccuracy of OOD detection when the available ID data is limited or biased, we design a global-local-aware scoring mechanism** that combines the Mahalanobis Distance [8] and maximum instance-wise cosine similarity to characterize ID data. The Mahalanobis Distance provides global awareness by approximating the ID data with a global Gaussian distribution, while the maximum instance-wise cosine similarity ensures local awareness by emphasizing instance-level relationships.
>
> Finally, we employ orthogonal regularization to facilitate the effectiveness and efficiency of our O3 framework when conducting continual unlearning. In fact, though applying LoRA can significantly reduce the computation and storage overhead during each unlearning operation, the accumulated LoRAs consume substantial resources as the unlearning requests increase. Therefore, we **enable the use of a single LoRA** rather than multiple LoRAs for continual unlearning and propose the **orthogonal regularization to disentangle every unlearning request in the LoRA parameter space, preventing the interference across requests**.

---

> ### Author Response · Authors · 2024-11-20
> **Concern about O3's robustness against attacks**
>
> > Concern about O3's robustness against attacks.
>
> We appreciate your insights on potential attack methods. To demonstrate the robustness of our approach against such attacks, we conducted experiments involving **adversarial attacks designed to bypass unlearning knowledge detection, membership inference attacks, and targeted relearning attacks**. Detailed descriptions and results of these experiments are provided in Appendix F.2 and Appendix E.4 of the main paper.
>
> In the real-world deployment of our O3 framework, there may be a concern that malicious attackers apply adversarial attacks to bypass unlearning knowledge detection. Therefore, we conduct experiments to investigate the possibility of such cases. Specifically, we implement an adversarial attack [9] against OOD detection that injects a certain perturbation to fool the OOD detector into identifying ID data as OOD data. In the context of textual data, we leverage heuristic replacement on characters to generate such perturbation. The experiments on TOFU show that the AUROC has no significant drop and the continual unlearning effectiveness remains nearly unchanged. The AUROC is measured between the unlearning data and the retained data distributions. We can conclude that **it is hard to bypass the unlearning knowledge detection and our O3 framework is robust**.
>
>
> |                 | SU   | DU   | AUROC |   | SU   | DU   | AUROC |   | SU   | DU   | AUROC |
> |-----------------|------|------|-------|---|------|------|-------|---|------|------|-------|
> | Ours w/ Attack  | 12.5 |   15 |  97.5 |   | 16.4 | 19.8 |  92.5 |   | 17.4 | 19.5 |  92.2 |
> | Ours w/o Attack | 12.5 | 14.4 |  97.8 |   | 15.8 | 20.3 |  92.5 |   | 15.5 | 19.7 |    93 |
>
>
>
> We also conducted Membership Inference Attacks (MIA) on the ScienceQA dataset following [2]. The training data for the pre-trained model contains the training data of the unlearning request, and the model can distinguish the unseen data in the test set from the unlearning request [10]. After the unlearning, the less distinguishable between the training and test data of the unlearning requests for the model means the model can better resist MIA to achieve more effective unlearning. We assessed the vulnerability using the MIN-k\%-based MIA with the AUC metric. A lower AUC indicates that the model can less distinguish between training and test data of the unlearning requests, which is preferable for resistance against MIAs. We compared O3 with SOPO, which was identified by Jia et al. [2] as superior to other baseline methods. As shown in Table below, **our method consistently outperformed SOPO**. For instance, at k=10, our method achieved an AUC of 0.559, which is lower than SOPO’s AUC of 0.655. Similarly, k=30/60, our AUC remained at 0.55, compared to SOPO’s AUC of 0.65.
>
>
> | k    | 5       | 10      | 20      | 30      | 40      | 50      | 60      |
> |------|---------|---------|---------|---------|---------|---------|---------|
> | SOPO | 0.673 | 0.655 | 0.652 | 0.652 | 0.652 | 0.653 | 0.653 |
> | Ours | 0.568 | 0.559 | 0.553 | 0.553 | 0.553 | 0.553 | 0.553 |
>
> Besides, we also conduct Targeted Relearning Attacks (thanks for Reviewer 9mDw's suggestion). To experiment on this, we followed the targeted relearning attack using public information setting described in [12]. Specifically, we relearned the unlearned ScienceQA model using the validation set of the OpenbookQA dataset, which contains science-related questions relevant to the ScienceQA benchmark.
>
> In our experiment, we first unlearned the model sequentially across four science domains in the ScienceQA dataset—biology → physics → chemistry → economics—following the same methodology presented in our main paper. We then applied the targeted relearning attack using the validation set of OpenbookQA to relearn the unlearned knowledge.
>
> We evaluated the performance of PO, SOPO, and our O3 framework before and after the relearning attack for the last unlearning requst, as shown in the table below. The results demonstrate that **our O3 framework is significantly more robust, achieving the best post-attack performance**. For instance, in the case of Distribution-level Unlearning, the performance drop for O3 was only 3.7, compared to 24 and 30.3 for PO and SOPO, respectively.
>
> |                | SU &darr;  | DU &darr;  | RD &uarr;   | CommonQA &uarr; | OpenbookQA &uarr; |
> |----------------|------|------|------|--------|----------|
> | PO           | 59.9 | 58.7 | 90.8 |   75.8 |     77.6 |
> | Relearned FOPO | 86.2 | 82.7 | 91.3 |   76.7 |     78.0 |
> |                |      |      |      |        |          |
> | SOPO           | 29.6 | 27.9 | 89.7 |   76.8 |     77.8 |
> | Relearned SOPO | 60.9 | 58.2 | 89.4 |   74.4 |     72.0 |
> |                |      |      |      |        |          |
> | O3             |  9.3 | 14.0 | 91.1 |   78.5 |     80.8 |
> | Relearned O3     | 15.5 | 17.7 | 89.6 |   75.0 |     72.6 |

---

> > ### Author Response · Authors · 2024-11-20
> > **Reference**
> >
> > [1] Liu, Sijia, et al. "Rethinking machine unlearning for large language models." arXiv preprint arXiv:2402.08787 (2024).
> >
> > [2] Jia, Jinghan, et al. "Soul: Unlocking the power of second-order optimization for llm unlearning." EMNLP (2024).
> >
> > [3] Yao, Yuanshun, Xiaojun Xu, and Yang Liu. "Large language model unlearning." arXiv preprint arXiv:2310.10683 (2023).
> >
> > [4] Zhang, Ruiqi, et al. "Negative preference optimization: From catastrophic collapse to effective unlearning." arXiv preprint arXiv:2404.05868 (2024).
> >
> > [5] Gu, Jia-Chen, et al. "Model editing harms general abilities of large language models: Regularization to the rescue." EMNLP (2024).
> >
> > [6] Gupta, Akshat, Anurag Rao, and Gopala Anumanchipalli. "Model editing at scale leads to gradual and catastrophic forgetting." arXiv preprint arXiv:2401.07453 (2024).
> >
> > [7] Gao, Tianyu, Xingcheng Yao, and Danqi Chen. "Simcse: Simple contrastive learning of sentence embeddings." EMNLP (2021).
> >
> > [8] De Maesschalck, Roy, Delphine Jouan-Rimbaud, and Désiré L. Massart. "The mahalanobis distance." Chemometrics and intelligent laboratory systems (2000).
> >
> > [9] Chen, Jiefeng, et al. "Robust out-of-distribution detection for neural networks." AAAI (2022).
> >
> > [10] Shi, Weijia, et al. "Detecting pretraining data from large language models." ICLR (2023).
> >
> > [11] Singh, Chandan, et al. "Rethinking interpretability in the era of large language models." arXiv preprint arXiv:2402.01761 (2024).
> >
> > [12] Hu, Shengyuan, et al. "Jogging the Memory of Unlearned Model Through Targeted Relearning Attack." arXiv preprint arXiv:2406.13356 (2024).

---

> ### Author Response · Authors · 2024-11-24
> **Kind Reminder before Reviewer-Author Discussion Phase Closure for Reviewer nwA1**
>
> Dear Reviewer nwA1,
>
> The conclusion of the discussion period is closing, and we eagerly await your response. We greatly appreciate your time and effort in reviewing this paper and helping us improve it.
>
> Thank you again for the detailed and constructive reviews. We hope our response is able to address your comments related to the difference between our work and regular approaches with exact unlearning, the unacceptance of retained data in practice, the technical novelty of O3, and the robustness of O3 against attacks. We take this as a great opportunity to improve our work and shall be grateful for any additional feedback you could give us. We fully understand that you may be busy at this time, but we hope that you could kindly have a quick look at our responses and assess whether they have addressed your concerns and warrant an update to the rating. We would also welcome any additional feedback and questions. We sincerely appreciate your dedication and time again.
>
> Best Regards,
>
> Authors of Paper 8760

---

> > ### Comment · Reviewer_nwA1 · 2024-11-26
> >
> > I thank the authors for their reply. These results partially resolve my question.
> >
> > The key purpose of machine unlearning is to remove the data from the existing trained model. Install a shell that may be able to block the access. This part is not convincing. Further details with attack evaluation, is ok to me. I will increase my rank to 6.

---

> > > ### Author Response · Authors · 2024-11-26
> > > **Thank you for your reply and Further question**
> > >
> > > > The key purpose of machine unlearning is to remove the data from the existing trained model. Install a shell that may be able to block the access.
> > >
> > > Thank you for your valuable feedback and for raising your rating! Regarding your concerns about the uniqueness of LLM unlearning, we’d like to share our perspective:
> > >
> > > First, we agree that machine unlearning generally aims to remove specific data from a model. However, **in the context of LLMs, additional considerations are necessary** to ensure unlearning is both reasonable and effective. The key distinction between LLMs and traditional ML models lies in their scale, which introduces two unique challenges: **1) Any updates to LLMs require substantial computational resources; 2) Such updates often result in uncertain and unpredictable changes to the model's utility**. To address these challenges, recent research has explored methods to partially or entirely avoid direct updates to LLMs during unlearning, focusing on reducing computational costs and mitigating utility loss.
> > >
> > > Additionally, from our perspective, **unlearning is not solely about the process—it’s the outcome that truly matters**. Specifically, effective unlearning should be reflected in the model’s inference behavior: knowledge that has been "unlearned" must result in distinct inference outputs compared to retained knowledge. In this regard, **our O3 approach aligns with traditional unlearning methodologies**.
> > >
> > > Lastly, **implementing access controls, such as filters or shells, to block knowledge as a means of unlearning is inherently ineffective**. As demonstrated in our main paper, even state-of-the-art OOD detection methods perform poorly in achieving accurately hard-label filtering in language. Furthermore, no reliable mechanisms currently exist for soft access control in LLMs that can be used for unlearning. Even if future advancements improve filtering accuracy, access blocking as a method of unlearning has a critical limitation: **it cannot generate appropriate responses for blocked queries (i.e., unlearned knowledge)**. Simple refusals or random outputs are insufficient, as various inference attacks—such as membership, attribute, and property attacks—can easily extract sensitive information related to the unlearned knowledge.
> > >
> > > In conclusion, we believe there are many exciting and important directions for improving LLM unlearning, and we hope this field continues to evolve. Thank you again for your constructive and insightful comments!

---

### Official Review · Reviewer_J9vg · 2024-11-04

**Soundness:** 3
**Presentation:** 3
**Contribution:** 3
**Rating:** 6
**Confidence:** 4

**Summary:**

The authors address significant practical challenges in developing machine unlearning techniques for large language models (LLMs), where current state-of-the-art approaches fall short due to their dependency on retained data and inability to manage continual unlearning requests effectively. To overcome these limitations, they propose the O^3 framework, which introduces an orthogonal low-rank adapter to enable continuous unlearning of requested data and an out-of-distribution detector to assess the similarity between incoming inputs and unlearning data. Comprehensive experiments show that O^3 achieves notably higher unlearning effectiveness and utility preservation than existing methods, all without relying on retained data, even under continuous unlearning conditions.

**Strengths:**

1. The authors solve the critical challenge of LLM unlearning by getting rid of the access to the retained data. The design of orthogonal LoRA demonstrates significant improvement in evaluation.

2. The authors conduct extensive experiments to evaluate the effectiveness of the proposed O^3 method.

**Weaknesses:**

1. During the inference, each testing instance x will be fed into all OOD detector backbones. This might limit the method's scalability when the unlearning requests increase due to the higher computational cost.

2. The experiments focus on the QA datasets, where each query only contains a single knowledge entity to be unlearned. The authors might need to evaluate the framework under more challenging and realistic settings, wherein for each query, there might be multiple knowledge entities to be unlearned. I am concerned about this because the OOD detectors are trained on single unlearning requests.

3. The authors should improve the scale of unlearning requests (more than just 4 requests) to validate the claimed contribution that the O^3 framework can handle continual unlearning settings.

**Questions:**

See weakness.

---

> ### Author Response · Authors · 2024-11-20
> **Concern about the computation cost and scalability**
>
> > Concern about the computation cost and scalability.
>
> We thank the reviewer for highlighting their concerns regarding the computational cost and scalability of our O3 framework. Below, we address these aspects and outline potential directions for future improvements.
>
> **Computation Cost**: As detailed in Appendix E.1 of our main paper, we measured the time overhead during the inference stage using the Calflops [1] tool with a single batch input. While **baseline methods require more resources during training**, they invoke the LLM during inference, registering 13,215 MFLOPS. In practical system deployment, regardless of the number of unlearning requests processed, our method enables Unlearning Knowledge Detection with OOD detector backbones to operate in parallel [2], consuming only 709.37 MFLOPS. Combined with the Soft-weighted Inference, which requires 13,255 MFLOPS, the total computational overhead is 13,964.37 MFLOPS—**representing just a 5.6% increase compared to the baselines**.
>
> Both our method and the baselines store the LLM, which occupies 12,862 MB. Additionally, our method introduces OOD-related storage (1,450 MB) and LoRA (39 MB), resulting in a total additional storage requirement of 11.6% over the baselines. Given that storage is typically more affordable and less resource-intensive compared to GPU usage, this overhead is unlikely to pose significant challenges in practical applications, especially for large companies.
>
> **Scalability**: As mentioned in the previous computation costs analysis, regardless of the number of unlearning requests processed, our method allows the parallel computation with 5.6\% higher computational cost which is reasonable and manageable for big organizations. **While our method incurs slightly higher overheads than the baselines, these costs bring about significant improvements of unlearning effectiveness and utility preservation, and they are amenable to significant reductions through future optimizations**. For example, replacing the separate embedding model for the OOD detector backbones with the LLM’s native embedding could reduce storage overheads by nearly 90% and computational consumption by approximately 95%.
>
> Specifically, in our implementation, the OOD detector backbone uses the encoder-only Roberta model. Although this model can extract high-quality representations, its performance is still limited when faced with very complex inputs compared to larger-scale language models. Therefore, we consider directly using the target LLM to detect unlearning knowledge in the future. This approach is feasible because, in the O3 framework, we use LoRA as an external module to achieve unlearning, and the original target LLM is available for inference. We should gain the following benefits if we replace the OOD detector backbone with an LLM. First, LLMs can better capture subtle text differences, improving OOD detection performance. Second, smaller language models like Roberta cannot effectively extract contextual information from complex and long contexts. Thus, if an unlearning request correlates the contextual information, such as the individual users' request to unlearn specific topics from their chat history with ChatGPT, Roberta-based OOD detection cannot achieve this. In contrast, LLMs can extract contextual information well [3], supporting more fine-grained OOD detection and more accurate ID data localization. Finally, using LLMs for OOD detection might eliminate the need for fine-tuning with ID data, as [4] suggested that LLMs could provide accurate OOD detection predictions for text classification without any fine-tuning. This could further improve our framework's efficiency.

---

> ### Author Response · Authors · 2024-11-20
> **Concern about whether O3 unlearns multi-entity knowledge**
>
> We thank you for suggesting a more realistic setting involving multiple knowledge entities to be unlearned per request.
>
> In response, we conducted experiments on the ScienceQA dataset, where we sequentially unlearned combinations of knowledge domains: biology and physics, followed by chemistry and economics. For each unlearning request, we mixed data samples from the two respective knowledge domains and followed the same continual unlearning process detailed in our paper for the ScienceQA dataset: (biology+physics)&rarr;(chemistry+economics). To evaluate the performance of our proposed O3 framework, we compared it with PO and SOPO, which were identified by Jia et al. [5] as superior to other baseline methods. As shown in the table below, our O3 framework significantly outperforms both baselines under this more complex scenario. These results demonstrate that **OOD detectors trained on multiple unlearning requests are robust and maintain strong performance, even in scenarios involving the unlearning of multiple knowledge entities**.
>
>
> |       | SU&darr; | DU&darr;   | RD&uarr;   | CommonQA&uarr;  | OpenbookQA&uarr;  |   | SU&darr;   | DU&darr;   | RD&uarr;   | CommonQA&uarr;  | OpenbookQA&uarr;  |
> |-------|------|------|------|--------|----------|---|------|------|------|--------|----------|
> | PO | 31.2 | 32.4 | 92.1 |   76.5 |     76.6 |   | 30.7 | 29.3 | 90.9 |   75.0 |     75.6 |
> | SOPO  | 27.9 | 28.7 | **91.9** |   77.1 |     79.8 |   | 23.5 | 23.1 | 91.1 |   76.0 |     77.0 |
> | O3    | **19.9** | **26.5** | 91.6 |   **78.2** |     **80.0** |   | **15.6** | **20.1** | **91.3** |   **78.2** |     **80.0** |
>
> Additionally, we would like to clarify that, **in our main experiments, a single unlearning request still encompasses multiple distinct knowledge entities.** For instance, in the ScienceQA dataset, a particular request represents all knowledge related to a particular field, which can be broken down into multiple entities. Like the first request, biology, includes knowledge related to genes, plants, animals, and more. Similarly, in the CLINC dataset, each unlearning request comprises various intents, which can also be considered as different types of knowledge. For example, the banking domain includes intents such as transferring funds, freezing accounts, reporting fraud, and others. Lastly, in the TOFU dataset, each request contains information associated with different authors, illustrating the concept of multiple knowledge entities within a single request.

---

> ### Author Response · Authors · 2024-11-20
> **Scale the unlearning with more requests**
>
> > Scale the unlearning with more requests
>
> Thank you so much for the suggestion of experimenting on more unlearning requests. We carried out experiments by dividing the TOFU-forget05 and TOFU-forget10 into 5 and 10 unlearning requests, respectively. In this way, each unlearning request contains information of 2 fictitious authors. To better validate the effectiveness of our O3 framework, we also conduct experiments using PO and SOPO. The detailed experiments are shown in tables below, from which we can observe that the **O3 framework substantially exceeds other baselines** in unlearning effectiveness and utility preservation. Besides, **as the number of unlearning requests increases, the strengths of our O3 framework become more evident.**
>
> **Unlearning 5 requests**
> | Request |  |  | 1 |  |  |  |  |  | 2 |  |  |  |  |  | 3 |  |  |  |  |  | 4 |  |  |  |  |  | 5 |  |  |
> |---|---|---|---|---|---|---|---|---|---|---|---|---|---|---|---|---|---|---|---|---|---|---|---|---|---|---|---|---|---|
> |  | SU&darr; | DU&darr; | RD&uarr; | RA&uarr; | WF&uarr; |  | SU&darr; | DU&darr; | RD&uarr; | RA&uarr; | WF&uarr; |  | SU&darr; | DU&darr; | RD&uarr; | RA&uarr; | WF&uarr; |  | SU&darr; | DU&darr; | RD&uarr; | RA&uarr; | WF&uarr; |  | SU&darr; | DU&darr; | RD&uarr; | RA&uarr; | WF&uarr; |
> | PO | 17.0 | 21.1 | 80.4 | 86.5 | 84.0 |  | 30.7 | 35.6 | 82.0 | 82.4 | 83.2 |  | 38.6 | 41.1 | 81.5 | 81.5 | 82.7 |  | 46.0 | 55.2 | 80.0 | 81.8 | 82.0 |  | 50.4 | 56.7 | 77.9 | 80.3 | 81.0 |
> | SOPO | 24.4 | 25.0 | 84.2 | 86.6 | 85.0 |  | 34.5 | 36.8 | 81.7 | 82.8 | 82.5 |  | 40.7 | 38.8 | 80.9 | 82.0 | 81.3 |  | 45.6 | 47.0 | 77.9 | 80.9 | 79.8 |  | 54.2 | 56.6 | 75.2 | 76.8 | 76.5 |
> | O3 | 13.0 | 14.5 | 85.7 | 89.0 | 86.3 |  | 14.2 | 14.0 | 85.5 | 89.0 | 86.0 |  | 16.2 | 17.5 | 85.5 | 88.8 | 86.3 |  | 16.5 | 18.4 | 85.4 | 88.6 | 86.2 |  | 17.0 | 19.2 | 85.2 | 88.8 | 86.0 |
>
> **Unlearning 10 requests**
> | Request |  |  | 1 |  |  |  |  |  | 2 |  |  |  |  |  | 3 |  |  |  |  |  | 4 |  |  |  |  |  | 5 |  |  |
> |---|---|---|---|---|---|---|---|---|---|---|---|---|---|---|---|---|---|---|---|---|---|---|---|---|---|---|---|---|---|
> |  | SU&darr; | DU&darr; | RD&uarr; | RA&uarr; | WF&uarr; |  | SU&darr; | DU&darr; | RD&uarr; | RA&uarr; | WF&uarr; |  | SU&darr; | DU&darr; | RD&uarr; | RA&uarr; | WF&uarr; |  | SU&darr; | DU&darr; | RD&uarr; | RA&uarr; | WF&uarr; |  | SU&darr; | DU&darr; | RD&uarr; | RA&uarr; | WF&uarr; |
> | PO | 20.4 | 25.7 | 81.5 | 86.5 | 84.8 |  | 33.2 | 34.0 | 81.0 | 85.7 | 83.8 |  | 42.2 | 44.8 | 80.3 | 84.0 | 81.4 |  | 48.9 | 50.7 | 78.8 | 81.8 | 80.5 |  | 50.5 | 52.7 | 77.9 | 80.6 | 80.2 |
> | SOPO | 27.7 | 31.6 | 83.7 | 86.5 | 84.3 |  | 32.8 | 35.5 | 82.0 | 82.9 | 83.7 |  | 40.2 | 41.7 | 80.9 | 82.4 | 81.0 |  | 47.8 | 47.0 | 80.4 | 81.5 | 80.5 |  | 50.6 | 54.6 | 78.7 | 79.5 | 79.8 |
> | O3 | 14.0 | 14.7 | 85.8 | 89.0 | 86.3 |  | 14.2 | 15.5 | 85.8 | 89.0 | 86.0 |  | 15.7 | 16.8 | 85.5 | 88.8 | 86.2 |  | 16.5 | 17.7 | 85.2 | 88.6 | 86.2 |  | 17.0 | 20.4 | 85.0 | 88.4 | 86.0 |
> | **Request** |  |  | 6 |  |  |  |  |  | 7 |  |  |  |  |  | 8 |  |  |  |  |  | 9 |  |  |  |  |  | 10 |  |  |
> | PO | 56.0 | 60.4 | 74.5 | 79.3 | 79.8 |  | 58.3 | 62.7 | 72.0 | 78.6 | 77.9 |  | 60.8 | 62.7 | 70.5 | 77.0 | 78.0 |  | 61.1 | 62.9 | 70.8 | 76.9 | 77.3 |  | 62.0 | 63.5 | 70.2 | 76.8 | 77.2 |
> | SOPO | 54.4 | 61.2 | 76.7 | 78.2 | 79.0 |  | 55.8 | 59.1 | 78.0 | 77.9 | 77.8 |  | 57.2 | 61.0 | 78.2 | 77.5 | 76.8 |  | 57.9 | 60.5 | 76.4 | 75.7 | 76.3 |  | 60.0 | 62.6 | 76.7 | 76.5 | 75.4 |
> | O3 | 18.4 | 20.8 | 85.0 | 88.2 | 86.0 |  | 20.4 | 22.5 | 84.8 | 88.0 | 85.7 |  | 20.7 | 24.0 | 85.0 | 88.0 | 85.5 |  | 20.0 | 22.3 | 85.2 | 88.2 | 86.0 |  | 23.4 | 25.0 | 85.0 | 87.5 | 85.5 |
>
> [1] https://github.com/MrYxJ/calculate-flops.pytorch
>
> [2] Agrawal, Amey, et al. "Vidur: A Large-Scale Simulation Framework For LLM Inference." MLSys (2024).
>
> [3] Ding, Yiran, et al. "Longrope: Extending llm context window beyond 2 million tokens." ICML (2024).
>
> [4] Uppaal, Rheeya, Junjie Hu, and Yixuan Li. "Is fine-tuning needed? pre-trained language models are near perfect for out-of-domain detection." ACL (2023).
>
> [5] Jia, Jinghan, et al. "Soul: Unlocking the power of second-order optimization for llm unlearning." EMNLP (2024).

---

> ### Author Response · Authors · 2024-11-24
> **Kind Reminder before Reviewer-Author Discussion Phase Closure for Reviewer J9vg**
>
> Dear Reviewer J9vg,
>
> Thank you for your initial constructive comments and insightful suggestions. We greatly appreciate your time and effort in reviewing this paper and helping us improve it.
>
> We hope our response is able to address your comments related to the computation costs, additional evaluation of multi-entity unlearning, and scalability analysis with larger number of requests. We take this as a great opportunity to improve our work and shall be grateful for any additional feedback you could give us. Your feedback is really important to us. We eagerly await any potential updates to your ratings, as they play a critical role in the assessment of our paper. Your thoughtful evaluation greatly aids in our paper's refinement and strength. We sincerely appreciate your dedication and time again.
>
> Best Regards,
>
> Authors of Paper 8760

---

> > ### Comment · Reviewer_J9vg · 2024-11-26
> > **Post-rebuttal Comments**
> >
> > Thanks for the authors' feedback. It has addressed my concerns, and I raise my rate of evaluation.

---

> > > ### Author Response · Authors · 2024-11-27
> > >
> > > Dear Reviewer J9vg,
> > >
> > > We sincerely appreciate your reply and rating raise! Thank you again for your constructive comments and insightful suggestions. We are so glad that your concerns have been addressed, which significantly enhanced the quality of our work. We would also appreciate any further feedback or questions.
> > >
> > > Best Regards,
> > >
> > > Authors of Paper 8760

---

### Official Review · Reviewer_9mDw · 2024-11-04

**Soundness:** 2
**Presentation:** 3
**Contribution:** 2
**Rating:** 8
**Confidence:** 3

**Summary:**

This paper presents a solution to the Continual Unlearning problem, in which the model provider attempts to continuously erase the influence of requested data. The authors motivate their problem formulation because existing model unlearning methods fail to account for the scenario where the unlearning requests emerge continuously, and the model provider lacks access to previous data. To this end, the authors propose a continual unlearning framework that includes LoRA adapters with orthogonal components to minimize the interference among different unlearning requests, along with an OoD detector that measures the similarity between input and unlearning data to decide on how unlearning LoRAs should be loaded.

**Strengths:**

1. This paper is well-written and motivated by real-world scenarios that require unlearning.
2. The design choices for the proposed unlearning pipeline are justified through ablation studies.

**Weaknesses:**

The authors partially motivate machine unlearning as “ a representative approach for model safety and security by removing the influence of undesired data on the target model.” I very much agree with the assertion. However, the evaluations of the proposed methods mainly focus on unlearning knowledge instead of unsafe behaviors. The usefulness of the proposed method could benefit from additional evaluations against safety-oriented unlearning benchmarks such as WMDP [1].

[1] Li, N., Pan, A., Gopal, A., Yue, S., Berrios, D., Gatti, A., Li, J.D., Dombrowski, A.K., Goel, S., Phan, L. and Mukobi, G., 2024. The wmdp benchmark: Measuring and reducing malicious use with unlearning. arXiv preprint arXiv:2403.03218.

**Questions:**

How robust is the proposed method against ”targeted relearning attacks”? For example, in a safety-critical scenario, can the supposedly unlearned knowledge be solicited again after finetuning the model on related public datasets? [2]

[2] Hu, S., Fu, Y., Wu, Z.S. and Smith, V., 2024. Jogging the Memory of Unlearned Model Through Targeted Relearning Attack. arXiv preprint arXiv:2406.13356.

---

> ### Author Response · Authors · 2024-11-20
> **Additional experiments of unlearning unsafe behaviors like those in benchmark WMDP**
>
> > Additional experiments of unlearning unsafe behaviors like those in benchmark WMDP.
>
> We appreciate the reviewers' insightful suggestion to include additional evaluations on unsafe behaviors. In response, we **have conducted experiments using the WMDP benchmark and a detoxification benchmark**.
>
> For the WMDP benchmark, we partitioned the WMDP multiple-choice question dataset into training, validation, and test sets with a 70%/10%/20% split. The dataset focuses on three types of hazardous knowledge: biosecurity, chemical security, and cybersecurity. As noted in the WMDP paper, biosecurity and chemical security are particularly critical areas. Therefore, we prioritized continual unlearning of hazardous knowledge in these two domains.
>
> Following SOUL [1], we also utilized LLaMA2-7b as the target model. To evaluate our proposed O3 framework, we compared its performance against PO and SOPO [1], which were identified by Jia et al. [1] as superior to other baseline methods. The results, summarized in the table below, **demonstrate that the O3 framework significantly outperforms these baselines in forgetting hazardous knowledge**. Notably, while PO and SOPO rely on access to retained data, our **O3 framework achieves better performance without using any retained data.**
> |      | SU &darr;   | DU &darr;   | RD &uarr;   |   | SU &darr;   | DU &darr;   | RD &uarr;  |
> |------|------|------|------|---|------|------|------|
> | PO | 45.3 | 33.9 | 55.0 |   | 20.4 | 24.8 | 55.9 |
> | SOPO | 30.0 | 26.4 | 52.5 |   | 24.8 | 21.5 | **57.2** |
> | O3   | **24.6** | **24.8** | **55.3** |   | **11.5** | **12.8** | **57.2** |
>
> In Appendix E.5 of the paper, we also present the performance of our proposed method on an unsafe behavior, LLM detoxification, which aims to prevent LLMs from generating toxic content. For this evaluation, we used negative samples from the training set of PKU-SafeRLHF [2], splitting them into three unlearning requests to simulate a continual unlearning scenario. Consistent with prior experiments and the methodology of SOUL [1], we adopted LLaMA2-7b as the target model.
>
> The unlearning effectiveness was assessed using the toxicity score (lower is better) on RealToxicity Prompts (RTP) [3] and PKU-SafeRLHF datasets. Utility preservation was measured by performance (higher is better) on the TruthfulQA benchmark [4]. As shown in the table below, **our O3 framework substantially outperforms the baseline methods PO and SOPO in unlearning effectiveness and utility preservation.** It is important to note that while PO and SOPO leverage access to sufficient retained data, **our O3 framework achieves superior performance without using any retained data**.
>
> |      | RTP&darr;    | PKU-SafeRLHF&darr; | TruthfulQA&uarr; |   | RTP&darr;    | PKU-SafeRLHF&darr; | TruthfulQA&uarr; |   | RTP&darr;    | PKU-SafeRLHF&darr; | TruthfulQA&uarr; |
> |------|--------|--------------|------------|---|--------|--------------|------------|---|--------|--------------|------------|
> | PO   | 0.0678 |        0.0830 |     0.2521 |   |  0.0670 |       0.0764 |     0.2522 |   | 0.0604 |       0.0711 |     0.2543 |
> | SOPO | 0.0675 |       0.0802 |     0.2537 |   | 0.0625 |       0.0766 |     0.2584 |   | 0.0567 |       0.0705 |     0.2644 |
> | Ours | **0.0569** |       **0.0605** |     **0.2563** |   | **0.0533** |       **0.0578** |     **0.2622** |   | **0.0495** |       **0.0462** |      **0.2650** |

---

> ### Author Response · Authors · 2024-11-20
> **Evaluation of O3's robustness against Targeted Relearning Attacks**
>
> > Evaluation of O3's robustness against Targeted Relearning Attacks.
>
> We appreciate the reviewers' suggestion to evaluate the robustness of our O3 framework under Targeted Relearning Attacks. To experiment on this, we followed the targeted relearning attack using public information setting described in [5]. Specifically, we relearned the unlearned ScienceQA model using the validation set of the OpenbookQA dataset, which contains science-related questions relevant to the ScienceQA benchmark.
>
> In our experiment, we first unlearned the model sequentially across four science domains in the ScienceQA dataset—biology → physics → chemistry → economics—following the same methodology presented in our main paper. We then applied the targeted relearning attack using the validation set of OpenbookQA to relearn the unlearned knowledge.
>
> We evaluated the performance of PO [1], SOPO [1], and our O3 framework before and after the relearning attack for the last unlearning requst, as shown in the table below. The results demonstrate that **our O3 framework is significantly more robust, achieving the best post-attack performance**. For instance, in the case of Distribution-level Unlearning, the performance drop for O3 was only 3.7, compared to 24 and 30.3 for PO and SOPO, respectively. We believe that the robustness against relearning is important and essential in real-world, and we plan to expore more in the future.
>
> |                | SU &darr;  | DU &darr;  | RD &uarr;   | CommonQA &uarr; | OpenbookQA &uarr; |
> |----------------|------|------|------|--------|----------|
> | PO           | 59.9 | 58.7 | 90.8 |   75.8 |     77.6 |
> | Relearned FOPO | 86.2 | 82.7 | 91.3 |   76.7 |     78.0 |
> |                |      |      |      |        |          |
> | SOPO           | 29.6 | 27.9 | 89.7 |   76.8 |     77.8 |
> | Relearned SOPO | 60.9 | 58.2 | 89.4 |   74.4 |     72.0 |
> |                |      |      |      |        |          |
> | O3             |  9.3 | 14.0 | 91.1 |   78.5 |     80.8 |
> | Relearned O3     | 15.5 | 17.7 | 89.6 |   75.0 |     72.6 |
>
> [1] Jia, Jinghan, et al. "Soul: Unlocking the power of second-order optimization for llm unlearning." EMNLP (2024).
>
> [2] Ji, Jiaming, et al. "Beavertails: Towards improved safety alignment of llm via a human-preference dataset." NeurIPS (2024).
>
> [3] Gehman, Samuel, et al. "Realtoxicityprompts: Evaluating neural toxic degeneration in language models." ACL (2020).
>
> [4] Lin, Stephanie, Jacob Hilton, and Owain Evans. "Truthfulqa: Measuring how models mimic human falsehoods." ACL (2021).
>
> [5] Hu, Shengyuan, et al. "Jogging the Memory of Unlearned Model Through Targeted Relearning Attack." arXiv preprint arXiv:2406.13356 (2024).

---

> > ### Comment · Reviewer_9mDw · 2024-11-24
> > **Thanks for the reply**
> >
> > I thank the authors for their reply. I will keep my current score.

---

> ### Author Response · Authors · 2024-11-24
> **Thanks to Reviewer 9mDw**
>
> Dear Reviewer 9mDw,
>
> We really appreciate your reply! Thank you again for your positive review and insightful suggestions. We are so glad that your concern could be addressed. We also appreciate any further feedback and questions.
>
> Best Regards,
>
> Authors of Paper 8760

---

### Author Response · Authors · 2024-11-20
**Response to all reviewers**

We would like to thank all the reviewers for their constructive comments and suggestions. In particular, we are sincerely grateful to Reviewers 9mDw, J9vg, and nwA1 for recognizing the motivation behind our work, which focuses on real-world LLM unlearning scenarios where the retained dataset may be unavailable, unlearning requests are continual, and LLM owners are reluctant to modify their base models due to significant computational costs and potential performance uncertainties. Besides, we also thank Reviewer 9mDw for their acknowledgment of the quality of our writing. Furthermore, we deeply appreciate the positive feedback from Reviewers 9mDw and J9vg regarding the extensive experiments conducted in our study, which convincingly demonstrate the effectiveness of the proposed method. Then we provide detailed responses for individual questions separately.

---

### Meta-Review · Area_Chair_Atig · 2024-12-18

**Metareview:**

This paper presents a solution to the Continual Unlearning problem, in which the model provider attempts to continuously erase the influence of requested data. The authors motivate their problem formulation because existing model unlearning methods fail to account for the scenario where the unlearning requests emerge continuously, and the model provider lacks access to previous data. To this end, the authors propose a continual unlearning framework that includes LoRA adapters with orthogonal components to minimize the interference among different unlearning requests, along with an OoD detector that measures the similarity between input and unlearning data to decide on how unlearning LoRAs should be loaded. This paper is well-written and motivated by real-world scenarios that require unlearning. The authors solve the critical challenge of LLM unlearning by getting rid of the access to the retained data. The design of orthogonal LoRA demonstrates significant improvement in evaluation, and the design choices for the proposed unlearning pipeline are justified through ablation studies. While the reviewers had some concerns about the evaluation, the authors did a particularly good job in their rebuttal. Therefore, all of us have agreed to accept this paper for publication! Please include the additional discussion in the next version.

**Additional Comments On Reviewer Discussion:**

Some reviewers raise the score after the rebuttal.

---

### Decision · Program_Chairs · 2025-01-22

Accept (Poster)